# Sensory input-dependent gain modulation of the optokinetic nystagmus by mid-infrared stimulation in pigeons

Tong Xiao[1,2†], Kaijie Wu[3†], Peiliang Wang[2,3,4†], Yali Ding[2], Xiao Yang[3], Chao Chang[3,5*], Yan Yang[1,2,6*]

[1]State Key Laboratory of Brain and Cognitive Science, Institute of Biophysics, Chinese Academy of Sciences, Beijing, China; [2]University of Chinese Academy of Sciences, Beijing, China; [3]Innovation Laboratory of Terahertz Biophysics, National Innovation Institute of Defense Technology, Beijing, China; [4]Key Laboratory of Electromagnetic Radiation and Sensing Technology, Aerospace Information Research Institute, Chinese Academy of sciences, Beijing, China; [5]School of Physics, Peking University, Beijing, China; [6]Institute of Artificial Intelligence, Hefei Comprehensive National Science Center, Hefei, China

*For correspondence:
changc@xjtu.edu.cn (CC);
yyang@ibp.ac.cn (YY)

†These authors contributed equally to this work

Competing interest: The authors declare that no competing interests exist.

**Abstract** Neuromodulation serves as a cornerstone for brain sciences and clinical applications. Recent reports suggest that mid-infrared stimulation (MIRS) causes non-thermal modulation of brain functions. Current understanding of its mechanism hampers the routine application of MIRS. Here, we examine how MIRS influences the sensorimotor transformation in awaking-behaving pigeons, from neuronal signals to behavior. We applied MIRS and electrical stimulation (ES) to the pretectal nucleus lentiformis mesencephali (nLM), an essential retinorecipient structure in the pretectum, and examined their influences on the optokinetic nystagmus, a visually guided eye movement. We found MIRS altered eye movements by modulating a specific gain depending on the strength of visual inputs, in a manner different than the effect of ES. Simultaneous extracellular recordings and stimulation showed that MIRS could either excite and inhibit the neuronal activity in the same pretectal neuron depending on its ongoing sensory responsiveness levels in awake-behaving animals. Computational simulations suggest that MIRS modulates the resonance of a carbonyl group of the potassium channel, critical to the action potential generation, altering neuronal responses to sensory inputs and as a consequence, guiding behavior. Our findings suggest that MIRS could be a promising approach toward modulating neuronal functions for brain research and treating neurological diseases.

## Editor's evaluation

This study will be of interest to systems neuroscientists considering neuromodulation techniques other than optogenetics or electrical stimulation. The work is important, as it provides new insights into the mechanisms and effects of mid-infrared stimulation (MIRS) on neuronal activity. Using optokinetic nystagmus in pigeons as a model circuit, it provides compelling evidence that depending on the cells' activity, MIRS can either increase or decrease neuronal firing – an effect that sets this technique apart.

## Introduction

Neuromodulation has historically been used both for circuit manipulation in neuroscience research and as a tool to treat patients with brain disorders. Scientific and clinical applications have co-evolved with each set of technological innovations. For example, deep brain and transcranial electromagnetic

stimulation directly alter neuronal excitability and connectivity, and their use in clinical settings is widespread (*Krauss et al., 2021*). Optogenetic stimulation has recently extended the toolkit to excite or inhibit specific groups of neurons selectively, but relies on genetic manipulations that currently limit its clinical application (*Deisseroth, 2015*). It will be of great significance to neuroscience research and clinical applications of neuromodulation if technological advances permit finer-resolution neuronal excitation and inhibition without genetic manipulation.

Optical infrared neuronal stimulation is emerging to serve this role due to its ability to deliver focused energy through tissue without direct contact. Initial studies have shown that near-infrared wavelength stimulation (NIRS) can excite neuronal responses *in vitro* (*Izzo et al., 2008*; *Albert et al., 2012*; *Shapiro et al., 2012*; *Entwisle et al., 2016*) and *in vivo* (*Wells et al., 2005*; *Wells et al., 2007a*; *Wells et al., 2007b*; *Richter et al., 2008*; *Xia et al., 2014*; *Xu et al., 2019*). Interestingly, there are also reports that NIRS can inhibit neuronal firing (*Cayce et al., 2011*; *Duke et al., 2012*; *Duke et al., 2013*; *Horváth et al., 2020*). Recently, there is evidence that mid-infrared stimulation (MIRS) with a specific wavelength can drive dramatic changes in neuronal firing rates and behavioral performance. MIRS can exert non-thermal effects on ion channels, and lead to gain modulation of action potentials based on current injections in the *vitro* slice preparation (*Liu et al., 2021*). MIRS can enhance spontaneous neuronal activities (*Zhang et al., 2021*) and sensory responses (*Tan et al., 2022*) in anesthetized animals. Consequently, prior reports show how MIRS accelerates associative learning in mice (*Zhang et al., 2021*), and regulates startle responses in larval zebrafish (*Liu et al., 2021*). Although our understanding of how MIRS neuromodulation impacts the brain is evolving, a convincing link between MIRS influence over *neuronal firing* and observed alternation of *behavioral performance* remains incomplete due to a lack of simultaneous stimulation and recording in awake-behaving animals. In this research, we aim to bridge the gap between the cellular targets of MIRS and its effects on a simple sensorimotor transformation in an awake-behaving pigeon.

Pigeons, as creatures of flight, highly depends on vision. While they fly or walk, the surrounding environment generates a large field of visual motion over the entire retina, known as optic flow (*Gibson, 1951*). In the presence of this visual motion, pigeons' eyes reflexively move to maintain stabilization of image on the retina, by ocular movements known as the optokinetic nystagmus (OKN). OKN occurs in two phases producing a characteristic irregular sawtooth waveform. A slow phase of pursuit eye movement closely tracks the moving field followed by a quick phase of ballistic saccadic eye movement that resets eyes to their primary position when they reach a maximal eccentricity (*Figure 1B*). In birds, the pretectal nucleus lentiformis mesencephali (nLM) is a crucial optokinetic nucleus responsible for encoding the horizontal optic flow and generating the OKN. The nLM is homologous to the nucleus of the optic tract and highly conserved across all vertebrates (*Fite, 1985*; *McKenna and Wallman, 1985*). The majority of nLM neurons prefer slow velocity and become excited by visual motion in the temporal-to-nasal direction, while other neurons are predominantly sensitive to the nasal-to-temporal, or vertical motion (*Winterson and Brauth, 1985*; *Wylie and Frost, 1996*; *Fu et al., 1998*; *Wylie and Crowder, 2000*; *Cao et al., 2004*; *Wylie et al., 2018*). Reversible inactivation or electrolytic lesions of the nLM impair generating horizontal OKN (*Fite et al., 1979*; *Gioanni et al., 1983*; *Yang et al., 2008a*; *Yang et al., 2008b*). Regarding connectivity, the nLM relays visual information from the retina directly or indirectly to the vestibular cerebellum, the oculomotor cerebellum, the thalamus, the brainstem, and other brain areas (*Figure 1D*). These connections are integral to the nLM's role in eye and head movements, and many other behaviors, including postural and steering control (*Cao et al., 2006*; *Yang et al., 2008b*; *Wylie, 2013*; *Ibbotson, 2017*; *Wylie et al., 2018*). This established circuitry and decades of work documenting these specific sensorimotor transformations the nLM performs make it an ideal candidate to study the effects of MIRS on behavioral and neuronal responses.

In this study, we applied MIRS in the pretectal nLM and studied several of its effects in awake-behaving pigeons. We found a reversible and gain regulation of pursuit velocity of OKN eye movements depending on the strength of visual inputs. Simultaneous recording of neuronal activity in the pretectal nLM and behavioral performance demonstrated that MIRS could facilitate or suppress firing activity depending on its ongoing sensory responsiveness levels. These modulations increased with the size of MIRS output power. Computational simulations suggest a candidate mechanism underlying these effects whereby MIRS preferentially enhances potassium permeability through K⁺channels,

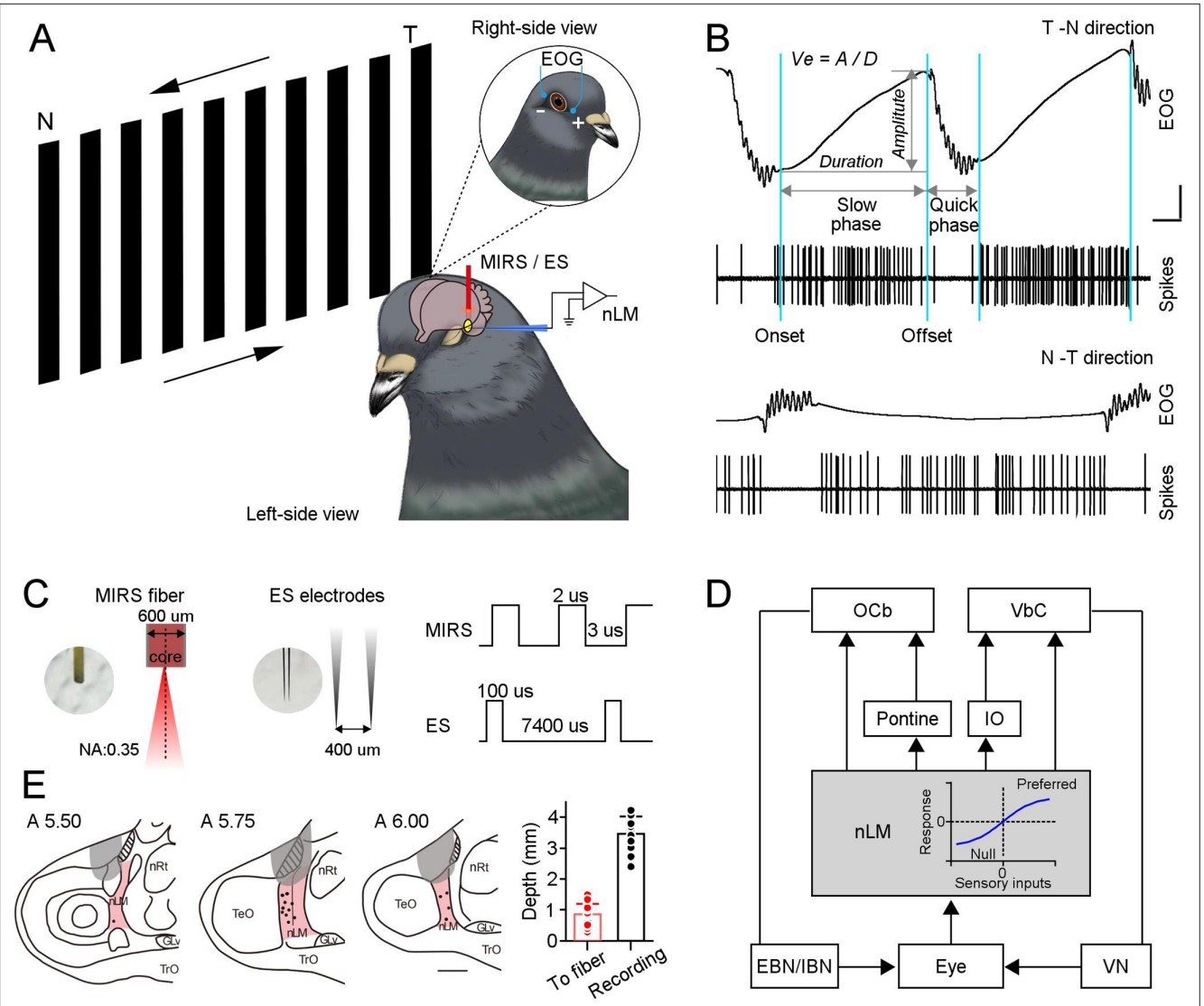

**Figure 1.** MIRS or ES applications to the pretectal nLM in pigeons. (**A**) Schematic drawing of neuronal activities (extracellular spikes) and behavioral performances (EOG) recording systems in awake-behaving pigeons, together with the arrangement of MIRS fiber or ES electrodes. Animals viewed grating motion in the T-N and N-T directions. (**B**) Raw traces of eye movements and action potentials of an example nLM while a pigeon viewing grating motion in the T-N (top traces) and N-T direction (bottom traces). Sky blue lines label time points of the onset and offset of pursuit eye movements during the slow phase of OKN. Arrow segments show the measurement of the amplitude and duration of the slow phase. An equation on the top defines pursuit eye velocity as amplitude divided by duration of pursuit eye movement. (**C**) The parameters of MIRS and ES used in the study. (**D**) The retina-nLM-cerebellum circuit involving the OKN eye movement generation in birds. nLM is a core sensory center that processes the direction and speed of visual motion in a large field and transfers these signals to the oculomotor system. The inserted plot in the gray box showed that firing responses of nLM evoked as a function of visual inputs. (**E**) Marked recording sites (dots) and MIRS/ES sites (gray shading) in the pretectal nLM (pink shading) across brain sections under study (from the interaural midpoint, A 5.5–6.0). On the right panel, the red bar presents the distance between the fiber tip and marked recording sites (n=15 pigeons, red dots). The black bar shows the depth of recorded neurons when recording electrodes were introduced laterally (n=37 neurons, black dots), as shown in A. Abbreviations are: OCb, oculomotor cerebellum; VbC, vestibular cerebellum; IO, inferior olive; nLM, the pretectal nucleus lentiformis mesencephali; EBN, excitatory burst neurons; IBN, inhibitory burst neurons; VN, vestibular nucleus; nRt, the nucleus rotundus; TeO, the optic tectum.

The online version of this article includes the following source data for figure 1:

**Source data 1.** Numerical data for *Figure 1B and E*.

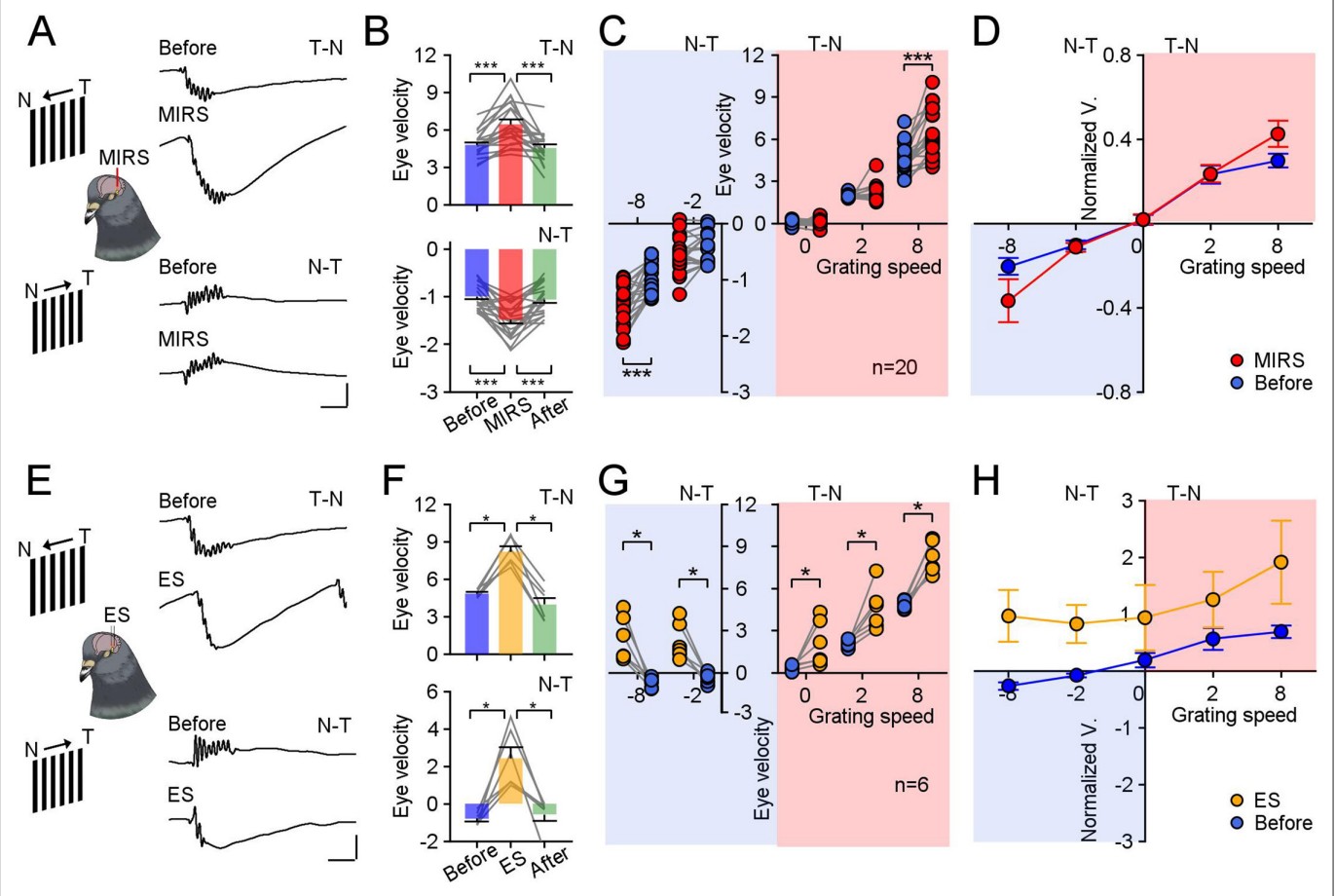

**Figure 2.** Comparison of modulation effects by MIRS and ES on OKN eye movements under different directions and velocities of grating motion. (**A and E**) Raw traces of OKN eye movements modulated by MIRS (**A**) and ES (**E**) while pigeons viewing grating motion in the T-N and N-T direction (top to bottom traces: before and during MIRS/ES in the T-N and N-T direction motion). Scale bars: 200ms, 4 deg. (**B and F**) Comparison of pursuit eye velocities to T-N and N-T grating motion (top and bottom plots) before, during, and after MIRS (**B**, n=20 pigeons) and ES (**F**, n=6 pigeons). (**C and G**) MIRS (**C**) and ES (**G**) modulated pursuit eye velocities under different grating motion stimuli (blue symbols: before stimulation; red symbols: during MIRS; yellow symbols: during ES). Two-sided Wilcoxon signed-rank test, * p<0.05, *** p<0.001. Gray lines represent data from individual animals. (**D and H**) Pursuit eye velocity data normalized to average values from the top 10 trials with highest peak pursuit velocities before MIRS (**D**) and ES (**H**) under different grating speeds. Error bars represent 1 SEM.

The online version of this article includes the following source data and figure supplement(s) for figure 2:

**Source data 1.** Numerical data for *Figure 2*.

**Figure supplement 1.** Modulation of sensorimotor behaviors by MIRS and ES in pigeons.

**Figure supplement 1—source data 1.** Numerical data for *Figure 2—figure supplement 1B-D and F-H*.

altering action potential generation in the nLM, which would modulate neuronal signals in brain network and guide sensorimotor responses.

## Results

### MIRS exerts gain modulation of pursuit depending on the strength of visual input

We introduced a large-field grating motion to pigeons and tracked their reflexive eye movements. Visual evoked OKN eye movements were recorded before, during, and after ~120 s of 10 mW MIRS applied in the left pretectal nucleus (*Figure 1A and C*) of 20 pigeons. In response to the temporal-to-nasal (T-N) grating motion of 8 deg/s (*Figure 1B* and *Figure 2A* top plots), they closely pursued moving gratings (*Figure 2B*, velocity: 4.78±0.22 deg/s; *Figure 2—figure supplement 1*, amplitude:

9.45±0.65 deg; duration: 2.23±0.22 s; mean ± SEM, n=20 pigeons) along the T-N direction, and then quickly saccade back to reset eye position. Once MIRS was turned on, animals significantly fasted their pursuit performance (velocity: 6.44±0.38 deg/s; n=20 pigeons, p=8.86×10$^{-5}$, two-sided Wilcoxon signed-rank test), but kept similar pursuit amplitudes and durations (amplitude: 10.12±0.70 deg; duration: 2.14±0.22 s; p=0.15 for amplitude, p=0.31 for duration, two-sided Wilcoxon signed-rank test). Conversely, when pigeons were introduced a grating motion of 8 deg/s in the nasal-to-temporal (N-T) direction (*Figure 1B* and *Figure 2A*, bottom plots), animals tracked grating motion with far less effective pursuit eye movements than ones in the T-N direction (*Figure 2B*, velocity: –0.99±0.06 deg/s; *Figure 2—figure supplement 1*, amplitude: –2.45±0.25 deg; duration: 2.96±0.28 s). When MIRS was turned on, animals again significantly fasted their eye movements to pursue in the N-T direction (velocity: –1.48±0.08 deg/s; n=20 pigeons, p=2.54×10$^{-4}$, two-sided Wilcoxon signed-rank test), without significant changes in pursuit amplitudes and durations (amplitude: –2.84±0.30 deg; duration: 2.73±0.25 s; p=0.07 for amplitude, p=0.06 for duration, two-sided Wilcoxon signed-rank test). There was asymmetry between the N-T and T-N OKN, which has been widely observed in lateral-eyed vertebrates (rabbits: *Collewijn, 1969*; pigeons: *Zolotilina et al., 1995*; rats: *Harvey et al., 1997*; mice: *Kodama and du Lac, 2016*). Although there were asymmetric OKN responses, MIRS significantly speeded up pursuit velocities in slow phases evoked by 8 deg/s grating motion along both T-N and N-T directions, in the individual and group animals.

Next, we conducted electrical stimulation experiments in 6 pigeons matching frequencies used in deep brain stimulation (*Hao et al., 2015*; *Mann et al., 2018*; *Valverde et al., 2020*). During the grating motion of 8 deg/s, a 120 s ES pulse train was applied similarly to our MIRS protocol (*Figure 1C* and *Figure 2E*). Unlike MIRS, ES can evoke pursuit eye movements towards the T-N direction, independent of the direction of grating motion (*Figure 2F*, top plot: T-N direction, before: 4.84±0.12 deg/s; ES: 8.19±0.42 deg/s; bottom plot: N-T direction, before: –0.80±0.14 deg/s; ES: 2.43±0.59 deg/s; n=6 pigeons, p=0.03 for both conditions, two-sided Wilcoxon signed-rank test).

To verify and explore these differences in the modulation of MIRS and ES, we further compared the effects of MIRS and ES on eye movements during the slow phase of OKN evoked by grating motion in multiple directions and speeds (*Figure 2C, D, G and H*). At a speed of 8 deg/s, MIRS significantly increased the component of pursuit eye velocity during the slow phase aligned with grating motion direction (*Figure 2B*). At a slower speed of 2 deg/s or in the absence of motion (0 deg/s), MIRS had a negligible impact on pursuit velocity (*Figure 2C*, 2 deg/s T-N direction: before: 1.98±0.04 deg/s, MIRS: 2.08±0.15 deg/s, n=16 pigeons, p=0.92; still grating: before: 0.01±0.06 deg/s, MIRS: 0.10±0.08 deg/s, n=11 pigeons, p=0.52; 2 deg/s N-T direction: before: –0.45±0.07 deg/s, MIRS: –0.56±0.08 deg/s, n=17 pigeons, p=0.12; two-sided Wilcoxon signed-rank test). The effect of ES on OKN was consistent across different visual conditions: ES deflected pursuit eye movements towards the T-N direction, regardless of directions or velocities of grating motion (*Figure 2F and G*, n=6 pigeons, 2 deg/s T-N direction: before: 2.10±0.10 deg/s, ES: 4.54±0.57 deg/s; still grating: before: 0.28±0.08 deg/s, ES: 2.10±0.59 deg/s; 2 deg/s N-T direction: before: –0.37±0.13 deg/s, ES: 2.24±0.49 deg/s; p=0.03 for all three visual conditions, two-sided Wilcoxon signed-rank test). To control for the possibility of an individual animal biasing our population analyses, we further examined pursuit velocity normalized to the top 10 non-stim trials with the highest pursuit velocity before MIRS/ES in each visual condition (*Figure 2D and H*). We compared the change of normalized data between before and during MIRS under different grating speeds. One-way ANOVA analyses revealed that gain modulation intensity of MIRS on pursuit eye movements during the slow phase depended on the strength of visual inputs ($F_{(4,79)}$=6.66, p=0.0001), in a manner different than the unidirectional modulation of ES ($F_{(4,25)}$=0.25, p=0.9094).

## MIRS excites and inhibits neuronal responses in the same pretectal neuron

To examine the effect of MIRS on neuronal responses, we performed extracellular recording and tested 43 nLM neurons in 15/20 awake-behaving pigeons before, during, and after MIRS stimulation. Relative to their spontaneous firing rate to still grating, over half of the recorded neurons (23/43) showed an increase in firing rate to motion in the T-N direction ("preferred direction"), with a decrease in the N-T direction ("null direction"). The other 20 cells showed the oppositive pattern of direction selectivity. All recorded nLM cells have been modulated by corollary discharge signals during

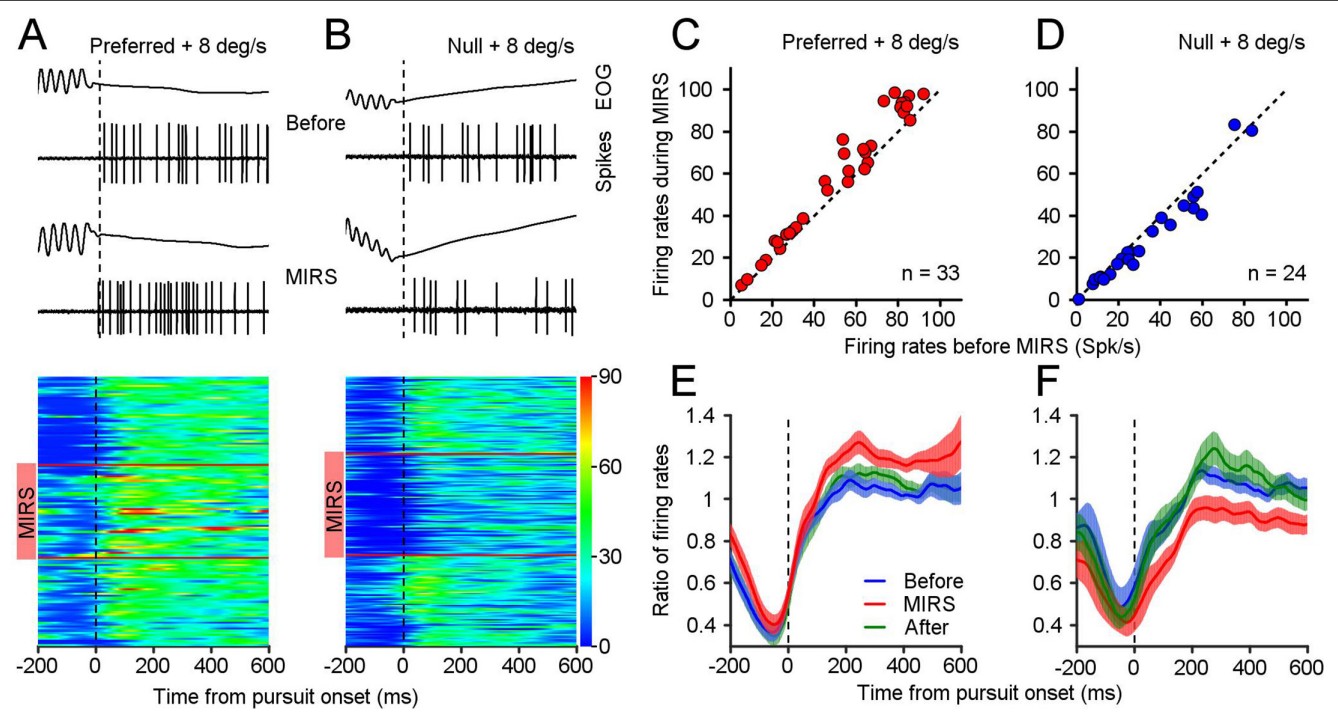

**Figure 3.** MIRS modulates visual responses of nLM neurons during the slow phase of OKN. (**A and B**) Representative eye movements and spiking responses to grating motion at 8 deg/s in the preferred (**A**) and null direction (**B**) of an example neuron. From top to bottom: example raw traces of EOG signals and action potential responses in a single OKN eye movement, and visual responses of the example neuron during individual OKN eye movements before, during, and after MIRS. Each horizontal colored band shows neuronal firing rates during pursuit of one OKN. Neuronal responses are color-coded with a scale on the right color bar (spikes/s). Red horizontal lines show the beginning and ending timepoints of MIRS. (**C and D**) Visual responses of population neurons showed the effect of MIRS in preferred (**C**, n=33 neurons) and null directions (**D**, n=24 neurons). Data points lie well above or below the dashed unity line. (**E and F**) Ratio of neuronal firing rates across population responses to grating motion at 8 deg/s before, during, and after MIRS in the preferred (**E**) and null (**F**) directions. Data were normalized to the mean value of visual responses in the 600ms interval after the onset of pursuit before stimulation. Error bars represent 1 SEM.

The online version of this article includes the following source data for figure 3:

**Source data 1.** Numerical data for *Figure 3*.

saccadic eye movements: their firing rates were inhibited during saccades, confirming prior report (*Yang et al., 2008a*). To investigate how nLM neurons code sensory information to guide pursuit eye movements of the OKN, we aligned the spiking activity to the initial of pursuit eye movements. The following data analysis focused on an interval from 0-600ms after the onset of pursuit. There are 37/43 neurons showed significant visual response changes in pursuit when MIRS at 10 mW applied (two-sided Wilcoxon signed-rank test, p<0.01). Among these 37 neurons, we tested 33 in their preferred directions, and 24 in their null directions. We focused on these 37 pretectal neurons with significant modulation for further data analysis.

*Figure 3* shows an example nLM neuron that prefers the N-T motion. At a grating motion of 8 deg/s in the neuron's preferred direction (N-T), visual responses reach an average of 30.99±6.84 spikes/s (mean ± SD, $n_1$=43 pursuits). Once MIRS was turned on, there is a potentiation of the visual response in that neuron to 37.68±11.35 spikes/s (*Figure 3A*, $n_2$=47 pursuits, $n_1$ vs. $n_2$: p=0.0034, two-sided Wilcoxon rank sum test), accompanied by a faster pursuit velocity toward the N-T direction (*Figure 2A*). By contrast, during motion in this neuron's null direction (T-N), neuronal firing rate fell to 26.25±6.73 spikes/s ($n_3$=60 pursuits). Interestingly, the MIRS further significantly enhanced neuronal inhibition to 18.40±6.87 spikes/s (*Figure 3B*, $n_4$=73 pursuits, $n_3$ vs. $n_4$: p=6.66×10$^{-10}$, two-sided Wilcoxon rank sum test), but sped up pursuit eye movements in the T-N direction (*Figure 2A*). In the given example nLM neuron, the same MIRS can increase and decrease neuronal firing rates to grating motion. These oppositive modulations on neuronal responsiveness were related to its levels

of visual responses in the preferred and null direction. This pattern was typical of most of the neurons we analyze below.

We compared each neuron's response to an 8 deg/s grating moving in their preferred and null directions before, during and after MIRS. All tested neurons had a greater firing rate during 600ms pursuit in their preferred direction (mean ± SD: 52.33±26.43 spikes/s, n=33 neurons) than their null direction (mean ± SD: 33.44±22.04 spikes/s, n=24 neurons) before stimulation. During MIRS, nearly all neurons showed a further potentiation of their preferred direction visual responses (*Figure 3C*, mean ± SD: 59.02±29.59 spikes/s, n=33 neurons, $p=1.76\times10^{-6}$, two-sided Wilcoxon signed-rank test), and a further inhibition in their null direction visual responses (*Figure 3D*, 29.16±21.23 spikes/s, n=24 neurons, $p=3.96\times10^{-4}$; two-sided Wilcoxon signed-rank test). We further calculated the ratio of firing rates by normalization to the mean value of visual responses in the 600ms pursuit before MIRS for individual cells, and then computed the average ratio across population as a function of time from the pursuit onset. Data showed that MIRS caused significant facilitation of neuronal excitation during the interval of 600ms of pursuit in the preferred direction (*Figure 3E*, n=33 neurons, before vs. during, $p=1.34\times10^{-6}$, two-sided Wilcoxon signed-rank test) and enhance neuronal inhibition in the null direction (*Figure 3F*, n=24 neurons, before vs. during, $p=1.29\times10^{-4}$; two-sided Wilcoxon signed-rank test). These bidirectional effects of MIRS occurred reversibly, which were linked with whether the nLM neuron was excited or inhibited by the sensory inputs.

## MIRS effects depended on the stimulus parameters

To tease apart interactions between MIRS and visual stimulation, we jointly varied visual and infrared stimulation parameters and examined how behavioral and neuronal responses changed. We examined how the strength of sensory input, the output power of MIRS, and the duration of MIRS affected responses (*Figure 4*).

First, we analyzed neuronal activity changes under different grating speed conditions when we applied MIRS at an output power of 10 mw (*Figure 4A*). We found a gradation firing rate in recorded nLM neurons by grating speed: as the grating motion increased from 2 to 8 deg/s in preferred or null (negative velocity) direction, average neuronal responses increased from ~45 to~52 spikes/s or decreased from ~40 to~33 spikes/s, consistent with the results of prior research (*Cao et al., 2004*). During MIRS, the increase of visual responses to the grating motion at 8 deg/s in the preferred direction was significantly larger than 2 deg/s in that direction (*Figure 4B*, mean ± SEM, 8 deg/s: 15.63% ± 1.72% ($n_1$=33 neurons); 2 deg/s: 5.25% ± 2.92% ($n_2$=13 neurons); $n_1$ vs. $n_2$ p=0.02, two-sided Wilcoxon rank sum test). Similarly, the decrease of visual activity to the 8 deg/s grating motion in the null direction was also significantly larger than 2 deg/s in the same direction (2 deg/s: –9.26% ± 2.66% ($n_3$=11 neurons); 8 deg/s: –17.53% ± 2.75% ($n_4$=24 neurons); $n_1$ vs. $n_3$, $p=2.10\times10^{-6}$; $n_1$ vs. $n_4$, $p=5.70\times10^{-10}$; two-sided Wilcoxon rank sum test). The percentage change of visual responses during MIRS was correlated with the grating motion condition for individual neurons (*Figure 4B*, each gray line presents an individual neuron).

Neurons in nLM can fire at different levels of spontaneous activities and evoke different visual responses based on the strength of sensory information. The MIRS effects on neuronal activity might relate to the ongoing neuronal firing rate (for example, the MIRS might enhance neurons' activities in the same way when these neurons fired at similar high frequencies by excitatory or inhibitory inputs). To confirm that the MIRS effect is indeed a manner of sensory inputs dependent, we have measured the effects under two well-controlled conditions. First, we checked the modulation while animals viewing still grating. nLM neurons fire at different 'spontaneous' firing rates without any excitation or inhibition by visual inputs of grating motion. During MIRS, these neuronal activities to still grating failed to show any significant changes (*Figure 4B*, mean ± SEM, –4.15% ± 3.87%, n=6 neurons, p=0.37, two-sided Wilcoxon rank sum test,). Second, we confirmed this was true for the subset of cells with similar evoked firing rates preceding MIRS but to grating motion in the preferred or null direction (*Figure 4C*). Group 1 preferred nLM cells reached a mean firing rate of 28.17±1.47 spike/s (mean ± SEM, n=7 neurons) during grating motion in their preferred direction. Group 2 null nLM cells reached a mean firing rate of 27.67±1.47 spike/s (n=7 neurons) for motion in their null direction. The firing rate range in each group was not statistically different from one another (p=0.80, two-sided Wilcoxon rank sum test), varying from ~22 to~36 spikes/s. When we compared MIRS effects on both groups, preferred cells showed a larger evoked response (p=0.02, two-sided Wilcoxon signed-rank test), while

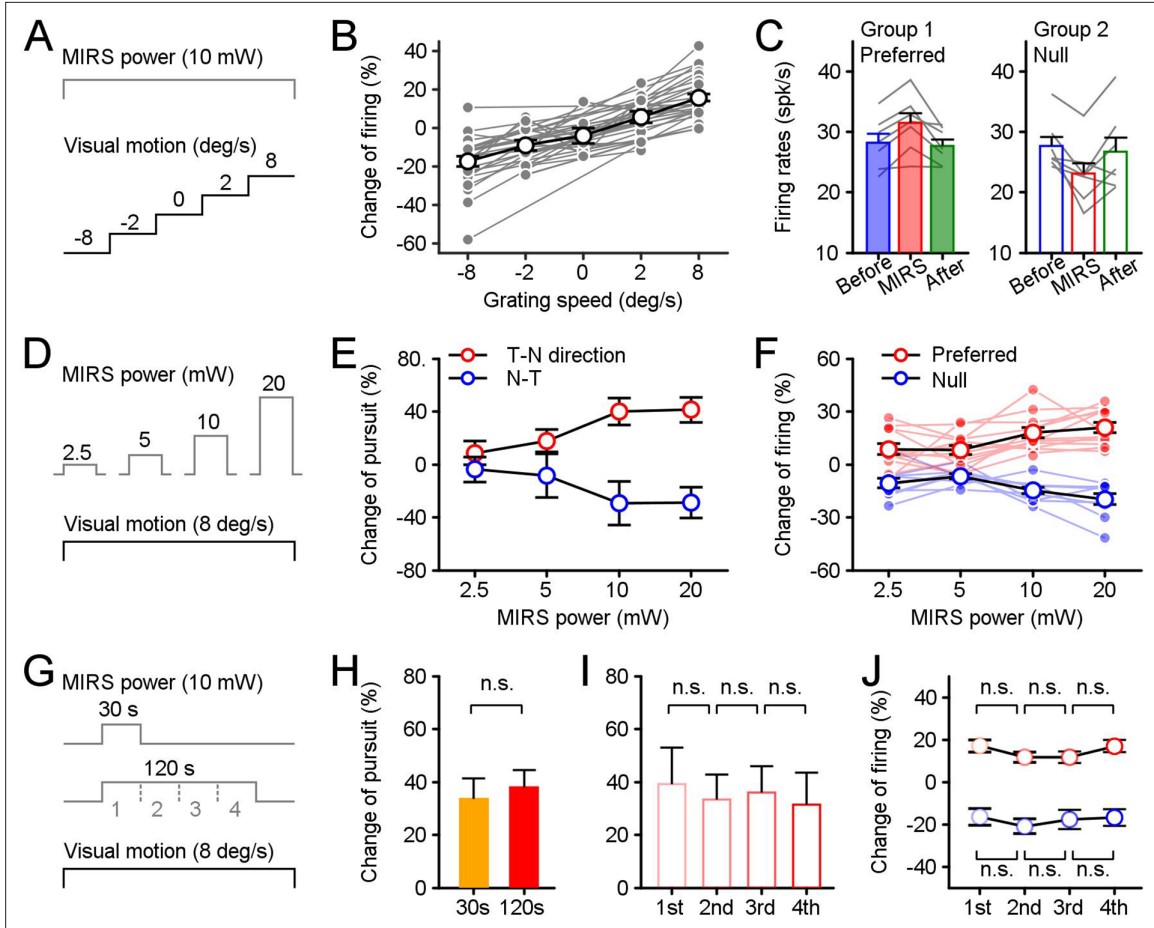

**Figure 4.** Effects of stimulus parameters on MIRS modulations of behavioral and neuronal responses. The parameters included the strength of visual inputs (**A**), the MIRS output power (**D**), and the duration of MIRS irradiation (**G**). (**A–C**) MIRS alters sensory coding in pretectal neurons associated with the level of visual responses. (**B**) The change ratio of visual responses across populations was correlated with the strength of visual inputs. (**C**) MIRS effects on different neuronal populations with similar evoked firing rates to grating motion at 8 deg/s in the preferred (Group 1, n=7 neurons) and null (Group 2, n=7 neurons) directions. (**D–F**) The size of MIRS output power alters modulations on behavioral and neuronal responses. (**E**) MIRS modulations on pursuit eye movements in the T-N and N-T directions increased with the MIRS output power. (**F**) MIRS effects on neuronal responses in the preferred (n=12 neurons) and null (n=9 neurons) directions increased with the MIRS output power. Filled symbols and lines show data from individual neurons. Open symbols with black line show the average across populations. (**G–J**) The duration of MIRS irradiation failed to modify modulations on behavioral and neuronal responses. (**H**) Comparison of 30 s (n=5 pigeons) and 120 s (n=20 pigeons) MIRS on pursuit eye velocities in the T-N direction. (**I** and **J**) Comparison effects on behavioral and neuronal responses in each 30 s period of 120 s MIRS irradiation. Pursuit velocity in the T-N direction (n=20 pigeons, **I**) and neuronal responses in the preferred (red cycles, n=33 neurons) and null (blue cycles, n=24 neurons) directions (**H**) failed to show significant changes across different periods. Two-sided Wilcoxon signed rank test, p>0.05. Error bars represent 1 SEM.

The online version of this article includes the following source data and figure supplement(s) for figure 4:

**Source data 1.** Numerical data for *Figure 4*.

**Figure supplement 1.** Profile of the mid-infrared light and the MIRS effects in other visual regions of pigeons.

**Figure supplement 1—source data 1.** Numerical data for *Figure 4—figure supplement 1*.

null cells showed more significant inhibition (p=0.02, two-sided Wilcoxon signed-rank test). These results have consistently supported that MIRS exerts gain modulation of neuronal signals, in a manner that is itself sensory responses dependent.

Next, we varied MIRS output power and examined its influence on pursuit velocity and nLM visual firing rates while the same grating motion was introduced to animals (*Figure 4D*). In response to the grating motion, pursuit eye velocities (*Figure 4E*, 10 pigeons) and neuronal responses (*Figure 4F*, 12 nLM neurons in 5 pigeons) scaled with MIRS output powers. When MIRS at a higher power applied, the percentage changes of pursuit eye velocity in the T-N and N-T direction were larger than ones when a lower power MIRS used (*Figure 4E*, red opened cycles for T-N direction: 41.32 ± 9.50% for

20 mW, 39.97 ± 9.98% for 10 mW, 17.81 ± 8.39% for 5 mW, and 8.68 ± 9.12% for 2.5 mW, mean ± SEM, n=19 MIRS times in 10 animals; blue opened cycles for N-T direction: 29.03 ± 11.68% for 20 mW, 29.53 ± 16.49% for 10 mW, 8.51 ± 16.59% for 5 mW, and 3.77 ± 9.42% for 2.5 mW, n=17 MIRS times in 10 animals). Although there were no significant changes in pursuit eye velocities between MIRS of 5 mW and a higher power of 10 or 20 mW (n=36 MIRS times in both directions in 10 pigeons, p=0.1620 for 5 vs. 10 mW or p=0.0573 for 5 vs. 20 mW, two-sided Wilcoxon signed rank test), the effects were significantly increased between MIRS of 2.5 mW and 10 or 20 mW (n=36 MIRS times, p=0.0057 for 2.5 vs. 10 mW or p=0.0022 for 2.5 vs. 20 mW, two-sided Wilcoxon signed rank test). A one-way ANOVA analysis further confirmed that the MIRS output power can effectively modify effects on eye movements in both the T-N and N-T directions at the $p<0.05$ for the four output powers ($F_{(3, 140)}$=3.4, p=0.0196). Meanwhile, in 12 nLM cells, we simultaneously recorded neuronal responses to grating motion at multiple output power levels (*Figure 4F*). Among them, we tested nine neurons in their preferred and null directions, and three additional neurons in only their preferred direction. When a higher power of 20 and 10 mW MIRS applied, neuronal responses were increased by 20.91 ± 2.78% for 20 mW and 17.92 ± 2.89% for 10 mW in the preferred direction (red dots and lines in *Figure 4F*, mean ± SEM, n=12 neurons) and decreased by 19.80 ± 3.10% for 20 mW and 14.66 ± 1.92% for 10 mW (blue dots and lines in *Figure 4F*, n=9 neurons). These modulations were significantly larger than changes of neuronal firing when a lower power of 5 and 2.5 mW MIRS used (the preferred direction: 8.21 ± 2.68% for 5 mW, and 8.61 ± 3.08% for 2.5 mW; the null direction: 6.75 ± 1.39% for 5 mW, and 10.62 ± 2.74% for 2.5 mW) (n=21 in the preferred and null directions, 5 vs. 20 mW: $p=9.22\times10^{-5}$, 2.5 vs. 20 mW: p=0.0026; 5 vs. 10 mW: $p=7.02\times10^{-4}$, 2.5 vs. 10 mW: p=0.0143, two-sided Wilcoxon signed-rank test). Like behavioral responses, MIRS power had a significant effect on evoked responses of the recorded nLM cells to grating motion stimuli (One-way ANOVA, $F_{(3,80)}$=7.92, p=0.0001).

To probe the effect of stimulation duration, we compared the influence of MIRS with 10 mW output power applied for 30 s vs. 120 s during 8 deg/s grating motion (*Figure 4G*). The evoked pursuit velocity did not vary substantially as a function of MIRS duration (*Figure 4H*, $n_{30s}$=5 MIRS times, $n_{120s}$=20 MIRS times, p=0.8651, two-sided Wilcoxon rank sum test). When we divided the 120 s duration condition into 4 sequential 30-s segments (*Figure 4I and J*), we again found that duration was not associated with a significant difference in eye velocity (n=20 MIRS times, One-way ANOVA, $F_{(3,76)}$=0.08, p=0.9698) or nLM firing rates (the preferred direction: $F_{(3,128)}$=1.18, p=0.3191; the null direction: $F_{(3,92)}$=0.26, p=0.8563, One-way ANOVA). Thus, the effect of MIRS was established in the first 30 s of stimulation and remained constant for 120 s.

In our study, the laser is coupled and emitted from optical fiber of 600 μm. The light spot diameter at a working distance of 850 μm (mean distance between fiber tip and recorded neurons, *Figure 1E*) can be about 1000 μm (*Figure 4—figure supplement 1A and B*). Thus the laser could have a good chance to irradiate the pretectal nLM with a width of ~1000 μm (*Figure 1E*). To further test the spatial specificity of MIRS, we measured the effect on eye movements when MIRS applied in other brain regions. We introduced the optical fiber to two adjacent visual nuclei located next to the nLM (*Figure 1E*), the optic tectum (TeO) and the thalamic nucleus rotundus (nRt), which are homologous to the superior colliculus and the pulvinar in mammals, respectively. The TeO is a principal destination of retinal ganglion axons and projects to the telencephalic entopallium via the nRt. The ascending tectofugal pathway is considered homologous to the colliculo-pulvinar-cortical pathway in mammals (*Benowitz and Karten, 1976*; *Wang and Frost, 1992*; *Hellmann and Güntürkün, 2001*; *Reiner et al., 2004*). Both nuclei are involved in visual information processing in birds, but not crucial for OKN generation. We run the same protocol as the main experiment, an 8 deg/s grating motion in the T-N direction and a 10 mW MIRS for 120 s. There were no significant changes in pursuit velocities during MIRS once the fiber applied in the OKN-irrelevant nuclei (*Figure 4—figure supplement 1C and D*, TeO: n=6 MIRS times in 3 pigeons, p=0.31; nRt: n=5 MIRS times in 3 pigeons, p=0.63; two-sided Wilcoxon signed-rank test).

## MIRS preferentially regulates the permeation of K⁺ channels

To explore the potential molecular mechanism underlying the gain modulation of neuronal responses by MIRS, we constructed models of voltage-gated K⁺ and Na⁺ channel subtypes. Recently, models of biomimetic channels have been widely used to investigate the translocation events of K⁺ and Na⁺ ions through ion channels (*Long et al., 2005*; *Zhang et al., 2012*). In our model (*Figure 5—figure*

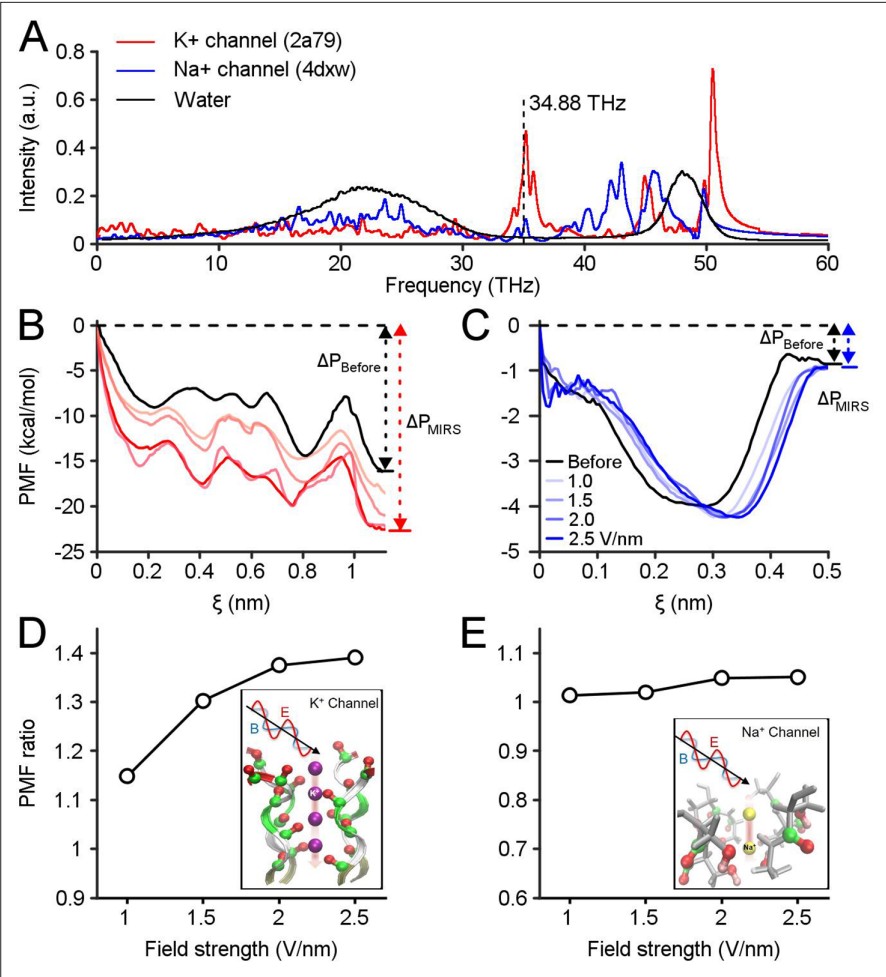

**Figure 5.** Computational simulations reveal preferential regulation of K⁺ channels by MIRS. (**A**) Absorption spectra of K⁺ and Na⁺ channels calculated using a molecular dynamics simulation. The vertical dashed line indicates MIRS with a frequency of 34.88 THz used in this study. (**B, C**) The potential of mean force (PMF) of K⁺ and Na⁺ ions permeate through ion channels before and during MIRS with a frequency of 34.88 THz, under field strengths varied from 0 to 2.5 V/nm (black and colored lines), respectively. (**D**) The PMF ratio of K⁺ channels increased by 1.15–1.4 folds at the exit site around $\xi$ =1.1 nm when the field strength increased from 1.0 to 2.5 V/nm. (**E**) The ratio of Na⁺ channels was kept closely to 1 at the exit site around $\xi$ =0.5 nm under different field strengths.

The online version of this article includes the following source data and figure supplement(s) for figure 5:

**Source data 1.** Numerical data for *Figure 5*.

**Figure supplement 1.** The composite atomic models contain the K⁺ channel (**A**) and the Na⁺ channel (**B**).

**Figure supplement 2.** The eigen-modes and intrinsic spectrum calculation for K⁺ and Na⁺ channels.

**Figure supplement 2—source data 1.** Numerical data for *Figure 5—figure supplement 2C*.

*supplement 1*), we constructed K⁺ channels containing the whole protein embedded in the middle of the phospholipid bilayer (DPPC molecules) to separate water and ions on each side. We defined the absorption spectrum of ion channels to MIRS based on the classical molecular dynamics (MD) method. The MD simulation showed two remarkable absorption fingerprint peaks of K⁺ channels located between 33–37 THz, and 50–55 THz (*Figure 5A*). Both fingerprint peaks are just out of the strong absorption spectral ranges of both water molecules (*Heyden et al., 2010*) and Na⁺ channels. The second absorption frequency of ~53 THz is consistent with prior research and well-studied (*Liu et al., 2021*). Note that the 34.88 THz simulation frequency we used in the initial results is closed to the maximum of the first peak of the K⁺ channels absorption spectrum, at least within half-height width of the fingerprint peak of 33–37 THz.

The protein structures of $K^+$ and $Na^+$ channels were tetramers, consisting of four chains and a narrow pore region (i.e. selectivity filter), that plays a decisive role in the permeation efficiency of $K^+$ and $Na^+$ ions. Using quantum chemistry calculations, we simplified the ion channel structure into a filter model. We further identified the specific absorption modes of ion channels during MIRS according to the filter structure extracted from the $K^+$ and $Na^+$ channels model. We calculated the intrinsic spectrum using Gaussian 09 code (*Frisch and Trucks, 2009*) based on density functional theory (DFT) at employing the B3LYP method and the 6-31G(d) basis set. We note that the absorption fingerprint peak of $-OH^-$ groups at the filter region of $Na^+$ channels are distant from ~34.88 THz (*Figure 5—figure supplement 2*), indicating that MIRS was mainly absorbed by $-C=O$ groups at the inner wall of the filter region of $K^+$ channels. Therefore, the low-frequency oscillation mode corresponding to MIRS with 34.88 THz could involve almost all the $-C=O$ groups (similar to $-C=O$ N-H) in the filter structure vibrating in an in-plane bending manner (*Figure 5A*). These data demonstrated that the effect of MIRS with 34.88 THz could enhance the resonance absorption of $K^+$ channels. While $-C=O$ groups at the inner wall of $K^+$ channels are frequency sensitive due to their collective resonance, these effects seldom occur in $Na^+$ channels or water molecules.

In response to a stimulus, the transfer of sodium ions through $Na^+$ channels and potassium ions through $K^+$ channels generates action potentials. To quantify the modulation of neuronal activity by MIRS, we computed the potential of mean force (PMF) as the change of free energy during ion permeability (*Bernèche and Roux, 2001*; *Li et al., 2021*). Action potentials are generated and released in milliseconds, with approximately $~10^7$ ions permeating across the membrane. These ions are passed sequentially through the filter region. Thus, the ion permeation can be studied on the level of single one (*Kopec et al., 2018*). Here we have simulated the permeability of a single ion, with an average permeation time on the order of nanosecond (*Kopec et al., 2018*) and different MIRS field strengths from 0~2.5 V/nm (*Zhu et al., 2020*; *Liu et al., 2021*). We used a steered molecular dynamics (SMD) method (*Abraham et al., 2015*) to analyze the ion permeation process. The PMF was sampled along the corresponding path and could reflect the permeation potential of individual ions along the filter region of ion channels (*Bernèche and Roux, 2001*; *Köpfer et al., 2014*). Our simulations suggest free energy changes between the entrance and the exit of the selectivity filter for both $K^+$ channels and $Na^+$ channels prior to MIRS (black lines in *Figure 5B and C*). It implied that potassium and sodium ions can permeate the selectivity filter with comparable energetic potential. The MD simulation with different field strengths showed that the potential energy changes of potassium ions through the filter became enlarged dramatically during MIRS (red lines in *Figure 5B*), while the energy of sodium ions only changed slightly through the filter configuration of $Na^+$ channels (blue lines in *Figure 5C*). To statically characterize the effect on ion permeability by MIRS, we defined a PMF ratio of $\Delta P_{MIRS}/\Delta P_{Before}$ ($\Delta P_{Before}$ and $\Delta P_{MIRS}$ represent before and during MIRS, respectively) for both sodium and potassium ions. In the simulation, MIRS increased the PMF ratio of potassium ions by 1.15–1.4 folds at the exit site of $K^+$ channels around $\xi = 1.1$ nm when the field strength increased from 1.0 to 2.5 V/nm (*Figure 5D*). At the same time, the ratio of $Na^+$ channels stayed near 1 at the exit site around $\xi = 0.5$ nm under different field strengths (*Figure 5E*). Together, our simulations demonstrate a mechanism by which the $K^+$ permeability can be selectively enhanced during MIRS with 34.88 THz. Increasing MIRS output power proportionally improves the efficiency of $K^+$ permeability, in a manner that only weakly modulates $Na^+$ channels. This enhancement of potassium permeability through $K^+$ channels will then lead to the changes in neuronal responses we document extensively above.

## Discussion

Our study provides evidence that MIRS can cause reversible and gain modulation on neuronal activity and sensorimotor behavior, by simultaneously recording neuronal and behavioral responses in awaking-behaving pigeons (*Figure 6*). The question of how MIRS modulates neuronal and behavioral responses were revealed in the greatest detail by directly controlling the strength of sensory input/ neuronal visual activity in our study. The effect of MIRS depended on the level of ongoing firing rates evoked by the direction and speed of visual motion, the MIRS output power, but not by the duration of MIRS. Computational simulations suggest a mechanism by which MIRS enhances $K^+$ permeance through the selectivity filter of potassium channels (*Figures 5 and 6B*) and causes the observed differences in neuronal responses. These findings suggest that MIRS with 34.88 THz could be an useful approach to modulate the neural circuits and motor responses in a sensory input-dependent manner.

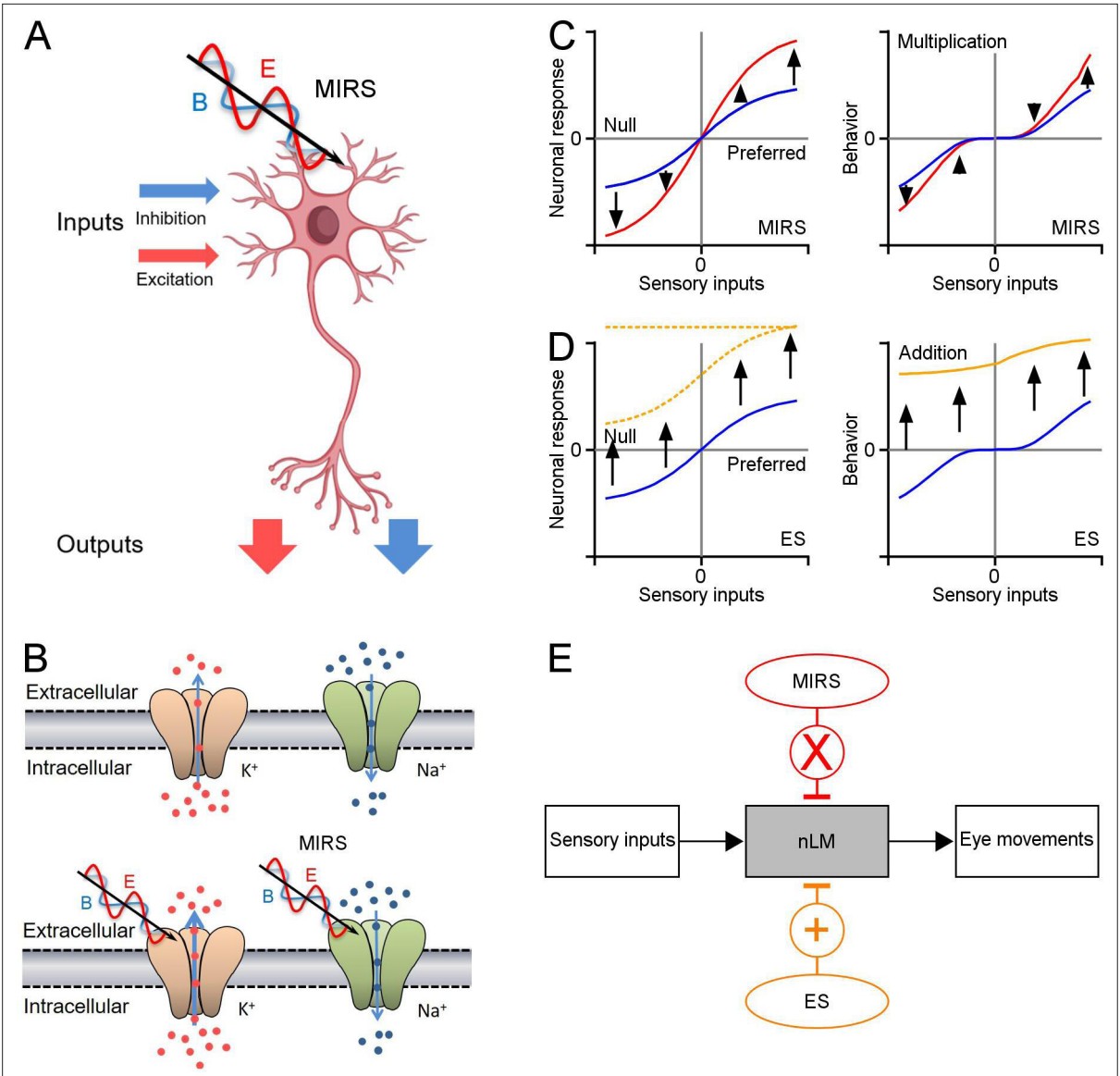

**Figure 6.** Schematic diagrams showing the mechanism of MIRS and ES suggested by our data. (**A**) MIRS produces gain modulations on visual responses in the same nLM neuron. (**B**) MIRS preferentially regulates the permeation of K$^+$ channels instead of Na$^+$ channels. (**C**) MIRS exerts multiplicative gain modulations on neuronal responses to sensory and behavior performance, suggested by our experimental data. Blue and red lines present neuronal and behavioral responses before and during MIRS. (**D**) ES could exert additive modulations on neuronal firing and cause unidirectional deflections in behaviour in our results. Blue and yellow lines present neuronal and behavioral responses before and during ES. Dashed lines present the possible changes in neuronal firing by ES suggested by prior research. (**E**) Summary of different effects of MIRS and ES on sensorimotor transformation: gain modulation by MIRS and additive modulation by ES.

The online version of this article includes the following source data and figure supplement(s) for figure 6:

**Figure supplement 1.** Extracellar recordings showed shortened action potential durations of pretectal nLM neurons during MIRS in the preferred (**A–D**) and null direction (**E–H**).

**Figure supplement 1—source data 1.** Numerical data for *Figure 6—figure supplement 1*.

We report that pursuit eye velocity during the slow phase of OKN covaries with the velocity of visual motion. We found that MIRS applied in nLM induced multiplicative gain modulations on pursuit eye movements based on the speed of visual motion (*Figure 6C*). MIRS can significantly speed up pursuit eye movements during the slow phase induced by a faster visual grating motion, but failed to significantly modulate pursuit when OKN was evoked by a slower visual grating motion. Note that if the grating was still (0 deg/s), pigeons only made spontaneous saccadic eye movements to search

surrounding without any slow pursuit eye movements. Under this visual condition, MIRS failed to initiate any pursuit-like eye movements in any directions. In contrast to MIRS, ES in the nLM caused a unidirectional defection of pursuit eye movements of OKN toward the T-N direction, independent of direction of grating motion (*Figure 6D*). Comparing behavioral evidence from MIRS and ES, we found that MIRS could induce gain modulations to regulate oculomotor behavior depending on the strength of sensory inputs.

Brain is constantly performing complex computations to encode sensory information and guide the behavioral performances. A nonlinear, multiplicative calculation by a single neuron could greatly enhance the computational power to process multiple incoming signals in the neuronal system (*Peña and Konishi, 2001*; *Gabbiani et al., 2002*; *Zhou et al., 2007*; *Groschner et al., 2022*). Our results showed that MIRS can cause reversible and multiplicative modulations on neuronal discharges depending on the ongoing levels of visual responses or the strength of sensory inputs (*Figure 4B* and *Figure 6C*). Consistent with prior reports, all recorded nLM neurons in this study showed tuning curves for the velocity of grating motion. Once the grating moved from a lower speed of 2 deg/s to a higher speed of 8 deg/s in both preferred and null directions, pretectal neurons could fire at different visual responsiveness levels, emitting neuronal discharges into low or high frequencies. MIRS can further inhibit or excite neuronal firing rates that occurred in the same neuron in a manner of its level of visual responses, respectively. Meanwhile, when animals viewed a given grating motion, the same MIRS could facilitate neuronal activity in one nLM neuron and suppress firing rates in the other neuron in a manner that the grating motion excited or inhibited neuronal discharges of that cell (*Figure 4C*). Therefore, MIRS can implement multiplication-like modulations on individual neurons in a sensory input-dependent manner.

A traditional explanation of these findings is that the MIRS could cause local thermal heat by water absorption, which changes capacitance of the transmembrane to excite cells (*Shapiro et al., 2012*), or activate thermosensitive TRPV channels to depolarize cells (*Albert et al., 2012*). By contrast, experimental evidence *in vitro* demonstrates that MIRS does not have a thermally mediated effect on ion channels and neuronal functions, especially when the distance was greater than 300 μm (*Liu et al., 2021*). MIRS could cause local thermal heat in our study. Meanwhile our evidence from behavioral and neurophysiological results is inconsistent with the thermal mechanism for several reasons. First, in our experiments, the distance between the MIRS fiber tip and recording sites was 848±351 μm (mean ± SD, *Figure 1E*), limiting thermal changes within a range of 2 °C (*Tan et al., 2022*). The MIRS effects on behavioral and neuronal responses were revisable after MIRS in our observations. Thus, it could be small chance that the limited thermal change during 120 s MIRS irradiation causes thermal damages in brain tissues. Second, MIRS's modulation of pursuit eye movements during the slow phase depends on the sensory inputs, which is not consistent with the thermal effect. If a mechanism of local thermal heat solely mediated the effect of MIRS, the gain of pursuit should be grating speed independent, contrary to our findings. Similarly, visual responses of the same recorded neuron could be excited or inhibited by MIRS depending on the direction of grating motion, and this enhancement of response scaled with the speed of grating motion. In summary, these findings suggest that thermally induced neuronal activity changes or vasodilation are insufficient to account for the results reported here.

The multiformity of ionic channels allows neurons to encode and transfer information by generating action potentials with a wide range of shapes, patterns and frequencies. This process must involve complex interactions with ion channels. Recent research in brain slice found that MIRS's multiplicative modulations of the action potential generation depended on the strength of current pules *in vitro*, and an increase in $K^+$ currents (not $Na^+$ currents) could lead to gain modulation (*Liu et al., 2021*). Consistent with previous reports, our ion channel simulations indicated that MIRS could cause the carbonyl group (-C=O) enriched on the $K^+$ channel selectivity filter to resonate, thereby decreasing membrane input resistance and increasing potassium ion flow. This mechanism may account for the modulation of neuronal responses to sensory inputs *in vivo* (*Figure 6B*). Potassium channels commonly play a significant part in membrane repolarization following action potentials. The increased potassium ion flow by MIRS could cause a faster and/or earlier repolarization and lead to a shortened action potential duration (*Figure 6—figure supplement 1*, $p<0.05$, two-sided Wilcoxon signed-rank test; *Liu et al., 2021*) and an enlarged afterhyperpolarization. As a result, stimuli that more strongly drive a neuron (i.e. motion in the preferred direction in our case) might cause faster recovery from the inactivation of sodium channels (*Bean, 2007*) and the prior action potential's refractory period. Then the neuron

can be facilitated to initiate a subsequent action potential, resulting in higher firing rates. With stimuli that weakly drive or inhibit the neuron (i.e. motion in the null direction), the raising K$^+$permeability might hinder the depolarization and retard the threshold potential of a subsequent action potential, resulting in lower firing rates. Although our computational simulations do not simulate every stage, from ion channel dynamics to the generation of action potentials, our data are consistent with an ionic mechanism underlying MIRS: MIRS could preferentially modify permeation of K$^+$channels, leading to alternations of action potential generation in a manner of ongoing firing levels depended on sensory inputs.

Contrary to the unidirectional behavioural and neuronal effects of electrical stimulation (*Figure 6D*), MIRS causes a sensory-specific gain modulation in visual responses of the same neuron and behavioral performance (*Figure 6C and E*). Unlike the optogenetic stimulation that would require the delivery of the transgenes encoding the light-responsive proteins, MIRS can selectively activate or inhibit neuronal responses by controlling the strength of sensory inputs in an individual or population cells, without genetic manipulation. These findings suggest that MIRS could be used as a promising neuro-modulation approach to excite and inhibit neuronal firing in the brain. There is the potential that MIRS could work for other higher cognitive functions. Recent research showed that MIRS could accelerate associative learning (*Zhang et al., 2021*) by stimulating the auditory cortex through opened or thinned intact skulls. At present, however, it is easier to interpret MIRS stimulation in sensory systems, where we have greater control over stimuli. Thus, methods to interpret MIRS applied to the associative areas serving higher cognitive functions are an appealing topic for future research. We hope that the presented details of MIRS effects on a simple sensorimotor transformation may lead to principles of MIRS that apply in other systems in the brain.

## Methods
### Animal preparation
We conducted experiments on 32 awake, behaving adult pigeons of either sex (*Columba livia*, body weight: 300–500 g). We randomly chose 20 animals among them for MIRS experiments and combined stimulation with recording in the pretectal nLM of 15/20. We randomly used six animals for electrical stimulation in the nLM experiments, and the left six animals for MIRS in the optic tectum (TeO) and the thalamic nucleus rotundus (nRt) experiments. We anesthetized each pigeon with an injection of ketamine (40 mg/kg) and xylazine (5 mg/kg) into the pectoral muscles. The anesthetic depth was assessed by breathing patterns and the toe pinch reflex (*Yang et al., 2008a*; *Yang et al., 2008b*; *Yang et al., 2017*). After placing the pigeon in a stereotaxic apparatus, we made an incision in its scalp and cemented a lightweight steel holder to the skull to provide head restraint in experiments. Based on the stereotaxic coordinates of the pigeon brain (*Karten and Hodos, 1967*), a recording window over the left tectum, and a stimulation window over the left caudal forebrain were surgically exposed by a dental drill. The overlying dura mater was untouched. After exposing these sites, we stitched the scalp over the window, treating wound sites with erythromycin ointment. Pigeons were returned to their home cages for recovery from anesthesia and received ibuprofen as an analgesic (20 mg/kg) for post-operative pain for several days after surgery. Within ~12 hr, pigeons' walking, pecking, and drinking behaviors had returned to normal. On an experimental day, we lightly anesthetized the pigeon with ketamine (4 mg/kg), wrapped it in a bag and, placed it on a foam couch. The head was stabilized to the stereotax using a rod connected to head holder. The wound edge and muscles were periodically infiltrated with lidocaine. The pigeon adapted to the restraint and sat unruffled on the couch. The left eye was covered, and the right eye was held open to visual stimuli during recordings, otherwise, its lid was allowed to move freely. Procedures followed with the guidelines for the care and use of animals established by the Society for Neuroscience and approved by the *Institutional Animal Administration Committee* at the Institute of Biophysics, Chinese Academy of Sciences.

### Visual conditions
We programmed and presented the visual stimulus in this experiment, a full-field square wave grating with a spatial frequency of 0.16 cycles/deg, using Matlab (Mathworks Inc, Matlab R2016b). The horizontal and vertical meridians of the visual field were rotated by 38° (*Britto et al., 1990*; *Fu et al., 1998*) to match pigeons' normal viewing conditions (*Erichsen et al., 1989*). The visual stimulation

was projected onto a large screen (130×140°) placed 40 cm away from the viewing eye by a projector (EPSON, CB-535W). The luminance of black and white stripes of grating was 0.1 and 6.6 cd/m². Animals produced spontaneous saccades when they viewed stationary gratings (0 deg/s) and OKN when they viewed moving gratings. In the experiments, the grating moved at 2 deg/s and 8 deg/s in the T-N and N-T directions to elicit OKN (*Gioanni et al., 1983*; *Yang et al., 2008b*).

## Mid-infrared stimulation and electrical stimulation

We applied MIRS using a quantum cascade mid-infrared laser (QCL, Institute of Semiconductors, Chinese Academy of Sciences) with a constant radiation wavelength of 8.6 µm, and frequency of 34.88 THz. The beam was collimated and matched with an infrared fiber coupling (polycrystalline Infra-Red fiber code: PIR600/700; core diameters: 600±15 µm; wavelength range: 3–17 µm; numerical aperture (NA): 0.35±0.05; type: multimode step index; temperature range was –50 to +80 °C). The pulse train of MIRS lasted 30 or 120 seconds with a pulse width of 2 µs, a repetition frequency of 200 kHz, and a duty cycle of 40% (*Figure 1C*). For MIRS experiments in the study, the average output power at the tip of fiber was 2.5, 5, 10, or 20 mW. The stability of output power over 120 s was better than ±0.5%, as measured by a MIR detector (NOVA II-3A, Ophir, Israel). We also measured profiles of the light spot when the mid-infrared laser passed through the coupler and optical fiber, and travelled 850 µm distance to the test surface. We used the MIR detector covered by a tinfoil with a 100 µm diameter hole to scan the intensity of the light, with a scanning resolution of 100 µm (*Figure 4—figure supplement 1A and B*). The spot diameter was ~1000 µm when the laser light intensity was reduced to $1/e^2$ of the peak in our setting. In most experiments, the fiber was inserted vertically through the window over the caudal forebrain to reach the left pretectal nLM (*Figure 1A and E*, from the interaural midpoint, A 5.75 mm; L 4.00 mm; H 5.5–6.0 mm). As a control, the fiber was introduced vertically through the forebrain window to reach the left TeO (*Figure 1E and A*, 6.0 mm; L 5.0 mm; H 5.5–6.0 mm), and the left nRt (*Figure 1E and A*, 6.0 mm; L 3.0 mm; H 6.5–7.0 mm) in six pigeons. There were ~2- to 10-min intervals between any two MIRS to allow recovery from the prior stimulation.

Electrical stimulation was generated using an isolated pulse stimulator (A-M Systems, model 2100) and applied for 120 s in the pretectal nLM, with parameters of 0.2 mA, 100 µs pulse width, 133 Hz (*Hao et al., 2015*; *Mann et al., 2018*; *Elias et al., 2021*; *Valverde et al., 2020*). The stimulation electrodes were glass-insulated tungsten bipolar electrodes with an exposed tip of 60 µm and a distance of 400–500 µm between the two tips (*Figure 1C*). The electrodes were advanced into the nLM following the same method as optical fibers in the MIRS (*Figure 1A*), with their poles in a mediolateral arrangement between L 3.5 mm to 4.5 mm.

## Data acquisition

### Electrooculogram recording (EOG)

Eye movements were recorded by an electrooculogram system (*Wohlschläger et al., 1993*; *Yang et al., 2008a*; *Yang et al., 2008b*). We inserted two EOG electrodes into the right orbital arch's anterior and posterior regions and placed a third electrode in the occipital bone as a reference. Eye position change signals were sampled at 2.5 kHz and stored simultaneously for off-line analysis together with the neuronal spike recordings (Cambridge electronic design limited, CED, power 1401). We smoothed EOG signals with a low-pass filter at a 5 Hz, and differentiated the signal to estimate eye velocity. For calibration purposes, we also imaged eye movements using an infrared video camera simultaneously with EOG recording (*Niu et al., 2006*; *Yang et al., 2008a*; *Yang et al., 2008b*). We defined the EOG electrode in the anterior region of right eye as the positive pole (nasal side), and the electrode in the posterior region as negative pole (temporal side). So the amplitude and velocity of eye movements, and the velocity of grating motion stimuli were represented as positive values in the T-N direction and negative for the N-T direction.

### Extracellular recording

We used homemade glass-insulated tungsten microelectrodes with an impedance of 1–3 MΩ for single-cell recording and electrolytic marking of recording sites (*Gioanni et al., 1983*; *Yang et al., 2008b*). To record the pretectal neurons in the MIRS light path, we placed the electrode laterally into the left tectum to target the nLM (from the interaural midpoint, A 5.5–6.0 mm, L, 4.0 mm; H, 5.5–6.0 mm). The depth range of recorded neurons was between 2.42 and 4.08 mm. The distance between recording

microelectrodes and fiber tips was about 848±351 μm (mean ± SD, n=15, *Figure 1E*). Extracellular action potentials were filtered using a bandpass filter of 300 Hz to 5 kHz (A-M Systems, Model 1800), and recordings were digitized at 25 kHz (CED, Power 1401 Cambridge electronic design limited). Spike sorting was performed offline (Spike2, Cambridge electronic design limited). Neurons in nLM had high levels of spontaneous activity, and were selective for the direction and velocity of grating motion. Based on this selectivity, we assigned a preferred and null direction to each cell.

At the end of each experiment, recording sites were marked by an electrolytic lesion (positive current of 30–40 μA for 20–30 s), and the animals were given a lethal dose of urethane (4 g/kg). The brain was extracted, postfixed for 24 hr (4% paraformaldehyde with 30% sucrose), immersed in 30% sucrose, and kept at 4 °C for 48 hr before being cut into 50 μm sections through the pretectum (Thermo Scientific Cryotome E). Using a standard light microscope, we reconstructed lesions, fiber tracts, and electrode tracts and verified that all recording sites were confined to the nLM (*Figure 1E*).

## Data analysis

For most of this study, we analyzed MIRS effects on the average eye movement velocity and firing rate in the 600ms interval following the onset of pursuit eye movements during the slow phase of OKN. The interval choice ensures that our measures are related to the visual response to grating motion during the slow phase of OKN. The onset and offset of the slow phase were determined by custom Matlab code (*Source code 1*) that detected the characteristic ~26 Hz oscillations in avian saccadic eye movements and then manually rechecked (*Figure 1B*). The amplitude and duration of the slow phase were measured from eye traces between the onset and offset of pursuit. Then the pursuit eye velocity during the slow phase was defined as amplitude divided by the duration of the slow phase in each OKN. We excluded trials where the slow phase was shorter than 600ms, although this was a negligible fraction of the tested conditions. Neuronal activities were collected before, during and after MIRS for about 120 s, aligned to the onset of pursuit, and smoothed by a Gaussian Kernel filter with an *h* value of 25ms. For each neuron, we computed average firing rate as a function of the 600 ms slow phase time (*Figure 3*), and compared its visual responses before and during MIRS. When firing rates were significantly increased or decreased through the slow phase of 600 ms during MIRS (p<0.01, n=600 ms, two-sided Wilcoxon signed-rank test), we reported neuronal firing of that neuron to be significantly facilitated or suppressed by MIRS.

Unless otherwise specified, if the data are paired, behavioral and neuronal responses were tested by a two-sided Wilcoxon signed-rank test. If they are two independent observations, an independent Wilcoxon rank sum test was used to compare data. Differences with p<0.05 were considered significant.

## Molecular dynamics simulation

The simulation was carried out to understand the power of ions permeation, and aimed to compare the permeability of $K^+$ and $Na^+$ ions during MIRS at nano-scale spatial and femto-second time resolution. The analyses were based on the eukaryotic model of voltage-gated $K^+$ channels (PDB ID: 2a79; *Long et al., 2005*) and $Na^+$ channels (PDB ID: 4dxw; *Zhang et al., 2012*), respectively. Our simulations used the force field of Charmm 36 and periodic boundary conditions (*Mackerell and Nilsson, 2008*). The connection element algorithm Ewald was used to deal with the electrostatic interaction (*Leeuw et al., 1980*). The Velocity-Verlet algorithm (*Andersen, 1983*) was performed to solve the motion equation, with the time step of 2 fs. All bond lengths were limited by the Lincs algorithm (*Swope et al., 1982*). In particular, for $K^+$ channels, the truncation of Lennard-Jones interaction and the real space part of Ewald sum were 1.90 nm, the convergence factor of Ewald sum was 1.65 nm, and the radius of K-space section was 10.4 nm. Meanwhile, for $Na^+$ channels, the truncation of Lennard-Jones interaction and the real space part of Ewald sum were 1.623 nm, Ewald and convergence factors were 1.65 nm, and the radius of K-space section was 10.4 nm. The process of the ion permeation events was divided into two stages according to before and during MIRS. First, the simulation system experiences temperature equilibration and pressure equilibration at 300 K room temperature to fully solvate mobile water and lipids around the protein to obtain the dynamical stable state. Then, we carried out a SMD process of ~1.5 ns for the $K^+$ ions permeation through the filter of $K^+$ channels, and ~0.75 ns of $Na^+$ ions permeation through the filter of $Na^+$ channels. We fixed the phospholipid bilayer and protein (except the filter region of $K^+$ and $Na^+$ channels) in the system to research the process of ion

permeation through the filter region under the NVT ensemble. To combine effects of the electromagnetic wave on the ion channel, MIRS with frequency of 34.88 THz was added into the whole simulation system (*Wu et al., 2020*). The intensity ratio of the electromagnetic component of an electromagnetic wave is equal to the speed of light. In the formula:

$$E(t) = A * u * cos(\omega t + \varphi) \tag{1}$$

where $A$ represented the amplitude of the electric field strength to generate the electromagnetic wave, varying from 0 to 2.5 V/nm in our simulations (*Zhu et al., 2020*; *Liu et al., 2021*), and $u$ and $\varphi$ represented the polarization direction and phase, thus set to (0, 0, 1) and 0 respectively. The electromagnetic wave frequency was computed as a function of the angular frequency $\omega$ by the equation $\gamma = \omega/2\pi$. In prior simulation research, action potentials were generated and released in milliseconds, with approximately ~$10^7$ ions permeating across the membrane, and ~10 ns permeation time for a single ion when there was no additional traction force (*Kopec et al., 2018*). Here we simulated the permeability of a single ion, with an additional traction force constant of 2000 kJ/mol/nm$^2$ applied. The permeation time required is set to ~1.5 ns for K$^+$ ions and 0.75 ns for Na$^+$ ions, at a pulling rate to 0.001 nm/ps. For the analysis of the ion permeation barrier, we used the steered molecular dynamics (SMD) method. The SMD produced a continuous conformation along the z-direction of the center line of ion channels. The PMF was sampled along the corresponding path.

We simulated the absorption spectrum of K$^+$ and Na$^+$ channels by using molecular dynamics methods based on the classical GROMACS code (*Abraham et al., 2015*; *Figure 5A*). The absorption spectra were calculated according to the Fourier transform of the velocity autocorrelation function of the total charge current of our simulation systems (*Heyden et al., 2010*). We set the time interval of spectrum sampling as 1 fs, and the total time of sampling as 50 ps. The absorption spectra were calculated based on the Fourier transform of the autocorrelation function of the total charge current (*Heyden et al., 2010*):

$$J(t) = \sum_i q_i v_i(t) \tag{2}$$

where $q_i$ represented the charge of the i-th atom, and $v_i(t)$ stood for the velocity of the ith atom at time $t$.

We further simulated the potential of mean force (PMF) as the change of free energy crossing configurations during ion permeability (*Bernèche and Roux, 2001*; *Li et al., 2021*; *Figure 5B and C*). Umbrella sampling (US) is a method that a series of initial configurations are sampled along a reaction coordinate defined between two groups; then, we simulated in the group of K$^+$/Na$^+$ harmonically restrained against the other fixed group via an umbrella biasing potential with the force constant of 2000 kJ/mol/nm$^2$ along the z-axis. Initially, a K$^+$ or Na$^+$ ion was placed in the z direction at the entrance of the selectivity filter. The Cl$^-$ ion was in line with the center of the protein channel. It was initially frozen and taken as the reference group. The initial configurations for the US simulations were extracted from the pulling process at approximately 0.2 Å (COM distance between the reference Cl$^-$ ion and the simulated K$^+$ or Na$^+$ ion) along the ion conductance path. During the US process, the protein (except for the key residues) was restrained. Thus, the shift of the whole protein due to system thermal and pressure fluctuations can be negligible. Based on the US data, the free energy profile was calculated with the WHAM method implemented in GROMACS code (*Abraham et al., 2015*).

The ion channel filter region is the critical factor determining an ion's permeability (*Kopec et al., 2018*). The selectivity filter of K$^+$ channels is composed of 24 residues (sequence index of 75~80; *Long et al., 2005*), and 12 residues for Na$^+$ channels (sequence index 9~11; *Zhang et al., 2012*). According to the filter structure extracted from the model of K$^+$ and Na$^+$ channels, the intrinsic spectrum was further calculated by using Gaussian 09 code (*Frisch and Trucks, 2009*) based on density functional theory (DFT) at B3LYP/6-31G(d) level (*Zhang et al., 2012*), thus to explore the specific absorption modes of an ion channel during MIRS.

## Acknowledgements

We thank Xi Xu, and members of our laboratory for helpful comments on an earlier version of the manuscript and discussions. Ramanujan Raghavan at New York University for help in editing the manuscript. Qian Wang, Chen Wu, and Haiyan Liu for invaluable technical assistance.

## Additional information

### Funding

| Funder | Grant reference number | Author |
| --- | --- | --- |
| Beijing Natural Science Foundation | Z210009 | Yan Yang |
| National Science and Technology Innovation 2030 Major Program | STI2030-Major Projects 2022ZD0204800 | Yan Yang |
| Chinese Academy of Sciences Key Project of Frontier Sciences | QYZDB-SSW-SMC019 | Yan Yang |
| National Natural Science Foundation of China | 32070987 | Yan Yang |
| Institute of Biophysics, Chinese Academy of Sciences | O1KF7208 | Yan Yang |
| XPLORER PRIZE No. 2020-1023 | | Chao Chang |
| National Natural Science Foundation of China | 12225511 | Chao Chang |
| National Natural Science Foundation of China | T2241002 | Chao Chang |

The funders had no role in study design, data collection and interpretation, or the decision to submit the work for publication.

### Author contributions

Tong Xiao, Data curation, Formal analysis, Investigation, Visualization, Methodology, Writing – original draft, Writing – review and editing; Kaijie Wu, Data curation, Software, Formal analysis, Investigation, Visualization, Methodology, Writing – original draft, Writing – review and editing; Peiliang Wang, Formal analysis, Investigation, Visualization, Methodology, Writing – original draft, Writing – review and editing; Yali Ding, Formal analysis, Investigation, Methodology; Xiao Yang, Formal analysis, Methodology, Writing – review and editing; Chao Chang, Conceptualization, Supervision, Funding acquisition, Investigation, Methodology, Writing – review and editing; Yan Yang, Conceptualization, Supervision, Funding acquisition, Investigation, Visualization, Methodology, Writing – original draft, Project administration, Writing – review and editing

### Author ORCIDs

Tong Xiao  http://orcid.org/0000-0001-8489-4731
Chao Chang  http://orcid.org/0000-0002-4081-1583
Yan Yang  http://orcid.org/0000-0003-3001-9178

### Ethics

Procedures in this study were in strict accordance with the guidelines for the care and use of animals established by the Society for Neuroscience. All of the animals were handled according to protocols approved by the Institutional Animal Administration Committee at the Institute of Biophysics, Chinese Academy of Sciences (#IBP-P-001(21)).

### Decision letter and Author response

Decision letter https://doi.org/10.7554/eLife.78729.sa1
Author response https://doi.org/10.7554/eLife.78729.sa2

## Additional files

### Supplementary files
• MDAR checklist

• Source code 1. Custom code to determine the onset and offset timepoints of the slow phases in OKN.

## Data availability

Source Data files have provided the numerical data used to generate the Figures 1 to 5, and the related figure supplement. Figure 6 in the current manuscript is the schematic diagram and Figure 5-figure supplement 1 is the atomic models. Source code file includes custom code cited.

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
