## [Editor Report]

This study will be of interest to systems neuroscientists considering neuromodulation techniques other than optogenetics or electrical stimulation. The work is important, as it provides new insights into the mechanisms and effects of mid-infrared stimulation (MIRS) on neuronal activity. Using optokinetic nystagmus in pigeons as a model circuit, it provides compelling evidence that depending on the cells' activity, MIRS can either increase or decrease neuronal firing – an effect that sets this technique apart.

---

## [Decision Letter]

**Decision letter after peer review:**

[Editors’ note: the authors submitted for reconsideration following the decision after peer review. What follows is the decision letter after the first round of review.]

Thank you for submitting the paper "Reversible and gain modulation of neuronal responses and sensorimotor behavior by mid-infrared stimulation" for consideration by *eLife*. Your article has been reviewed by 3 peer reviewers, and the evaluation has been overseen by a Reviewing Editor and a Senior Editor. The reviewers have opted to remain anonymous.

Comments to the Authors:

We are sorry to say that, after consultation with the reviewers, we have decided that this work will not be considered further for publication by *eLife*.

Specifically, the reviewers agreed that the results are potentially interesting to a broad audience, but there were too many questions about the stimulation protocol and its effect on neuronal activity, which undermined the conclusions of the study. It was felt that the manuscript would require major reworking, and likely additional experiments, to appropriately characterize the impact of MIRS stimulation and determine the reliability of its effects on neuronal activity and behavior.

*Reviewer #1 (Recommendations for the authors):*

Recent studies suggest that midinfrared stimulation (MIRS) could be a potential neuromodulation strategy, which exerts nonthermal and reversible effect on neuronal signaling and animal behavior. However, there is no direct evidence showing the effect of MIRS on neuronal firing in awake behaving animals. In this study, the authors examined the alternations of neuronal spiking activities in response to MIRS and the association between these alternations with behavioral performance (eye movement in pigeon) in vivo, and compared the modulatory effects of MIRS with electrical stimulation (similar to that with high-frequency deep brain stimulation). With simultaneous extracellular single-unit recording and electrooculogram recording, the authors demonstrated that MIRS with a wavelength of 8.6 μm produced reversible gain modulation on neuronal responsiveness and behavioral performance. They further dissected the underlying potential mechanism using molecular dynamics (MD) simulations and showed that MIRS with specific wavelength could cause carbonyl groups (-C=O) at the ion selectivity filter of K^+^ channel to resonate with the mid-infrared light, leading to the gain modulation of neuronal responsiveness and changes in eye movement. Overall, the electrophysiological results are of high quality and provide strong evidence showing the effect of MIRS on neuronal signaling and sensorimotor behavior. The MD results also provide insight into molecular mechanisms (K^+^ channels) underlying these modulatory effects. However, a weakness of the manuscript is that the methods and results are not well described.

1. Please provide detailed information about MIRS laser source and its application. There is no clear description about MIRS parameters. The type of infrared fiber is IRF-S-9 (the diameter should be 9 μm), but the diameter is described as 600 μm.

2. Please provide the pulse width and repetition frequency of MIRS. The specific parameters (including continue wave or pulse train) should be provided clearly, these parameters will all impact the MIRS effect.

3. It would be nice to have a schematic drawing of the experimental configurations, particularly the relative position of the MIRS fiber and the recorded brain region. The authors stated that the distance between the MIRS fiber tip and the electrode recording site was kept ~850 µm. The fiber was inserted vertically (what's the depth?) into the brain tissue. Was the nLM region in the light path?

4. What does the "80 mW" represent? the average power? The peak power? Had the authors measured the real output power at the tip of MIRS fiber? What is the estimated power density at the recorded brain region? At least, the power density in air should be given. What was the temperature rise at the fiber tip? Was there any tissue damage? It would help if the authors provided histochemical examination of the brain tissues before and after MIRS, including the nLM region and the fiber tip region. What are the changes in neuronal spiking activity near the fiber tip < 100 μm?

5. Why was the pursuit velocity (but not the duration and distance) altered by MIRS? What's the corresponding relationship between the firing frequency of nLM neurons and the eye movement parameters? How was the pursuit eye velocity analyzed? Please describe these parameters (velocity, duration and distance) and provide details about their measurements. Moreover, the authors performed data analysis of nLM neuron spiking activities within a time window from 0-600 ms after the onset of pursuit. It should be stated clearly in Figure 1 where the onset of pursuit was. The baseline firing activity without visual stimulation should be measured. Were there any effects on baseline activity upon MIRS application? From Figure 3C, it is clear that the firing frequency during the rapid eye resetting phase (i.e. 300 ms before the onset of pursuit) was altered with MIRS. In Figure 3E, the MIRS effect was not reversible? Spiking activity during the rapid phase (before the pursuit onset) was still high when the MIRS was turned off.

6. What do the gray bars in Figure 1. D-F stand for?

7. There is inconsistency between results description and Figure 2A. No significant difference between MIRS and Ctrl group?

8. The statistic results are inaccurate. Are they significantly different in each condition?

9. Should the baselines be aligned in Figure 4A-B? Please describe the statistical results in detail (Paired or independent Student's t-test, repeated-measures ANOVA).

10. The measurement of the extracellular spike waveforms in Figure S3 is not appropriate. The duration between the trough and the peak (i.e. peak to peak duration) should be used.

11. "Finger" in this manuscript should be changed to "Fingerprint peak".

12. From a chemical point of view, the -C=O group only has an intrinsic stretching vibration mode between carbon and oxygen atoms at a frequency of ~53 THz (not 34.88 THz). Although there is an absorption peak of the group at 34.88 THz calculated from the MD simulation, the source of this peak should be explained.

13. In this study, the total time of MD simulations is ~10 ns, which is too short for a biological system. It is important to confirm that the MD results are independent of the simulation time.

14. The field strength applied is 2.5 V/nm, too strong? Is this strength critical for the gain modulation?

15. The filter region of ion channel should be "selectivity filter", not "selective filter".

16. PMF is the key data to indirectly support the conclusion that MIRS could preferentially enhance potassium permeability through K^+^ channels to alter action potential generation, but it is not a direct measurement of the change in ion flux. It is important to confirm whether changes in PMF have an effect on K^+^ and Na^+^ currents. What is the direct effect of MIRS on ion currents?

17. Can the authors explain the physical mechanisms for the rightward shift of PMF for Na^+^ channels.

18. What is the dependence of PMF and ion currents on MIRS intensity?

19. What is the direction of the mid-infrared light? From Figure 5B and C, the direction is parallel to channel pores?

*Reviewer #2 (Recommendations for the authors):*

In the manuscript Xiao et al. investigate the impact of mid-infrared stimulation (MIRS) on neuronal activity in awake behaving pigeons. Earlier studies have found that MIRS can excite neuronal responses, but other studies also found inhibition of neuronal firing. In their paper, Xiao et al. investigate the mechanism of MIRS that might explain these diverse findings as reported in the literature so far. They investigated the impact of MIRS and electrical stimulation on neuronal signals as well as on the behavioral performance of pigeons during visually guided eye movements. They found that, unlike electrical stimulation, MIRS rather resulted in gain modulation of firing activity. Therefore, MIRS increased firing rates of highly active neurons, while it reduced the firing of less active cells. Likewise, pursuit eye movements were facilitated by MIRS. Moreover, Xiao et al. explained their experimental findings using computational methods to simulate molecular effects which suggested enhanced permeability of potassium channels.

The manuscript from Xiao et al. combines different methods including neurophysiology, behavior and computational modelling to comprehensively investigate the mechanism behind MIRS. Their electrophysiological findings are congruent to their behavioral results supporting the idea that MIRS results in gain modulation. Although this study was performed in pigeons, the findings could be of interest not just for avian researchers but for the whole field of neuroscience. This is the case as they shed light on the mechanisms of MIRS, so that it can be used more purposefully in future as their findings suggest that it is not inhibitory or excitatory per se but results in gain modulation.

However, the introduction and/or discussion is missing a section explaining why the effects of MIRS were investigated in pigeons. While the use of diverse model organisms is desirable, the authors motivate their study primarily with clinical applications and say that human research could benefit from MIRS. It is so far not completely clear why the model organism and the specific brain area were chosen in this study. To determine the generalizability of their findings, control experiments in other brain regions or species would be desirable.

Moreover, the authors say that in the clinical setting there is the need for neuromodulation methods that, without genetic manipulation, have the potential for neuronal inhibition and excitation. In parts of the manuscript the authors create the impression that this is possible with MIRS. However, based on their findings, the effect of MIRS is not directly controllable, as it varies with the strength of sensory input/neuronal activity. In their experiment the authors were able to control the sensory input into the cell population by varying the visual gratings. However, this is easy for sensory systems and becomes more complicated in higher associative areas. How well could the effect of MIRS be controlled in these areas?

The method section is missing important information. For example, the statistics and surgical procedures are not explained. In some other sections, the authors refer to other publications, which makes it hard for the reader to fully understand or assess the procedures. The stimulation protocol is also a cause for concern. Testing different stimulation protocols and their effects on neuronal activity would be necessary to fully characterize the effects of MIRS. So far only one protocol was used and it is not made clear why.

In the Results section it is sometimes hard to follow the statistical procedures. It is not always indicated what the number of subjects was, which test was performed and why, and whether a within or between subjects design was applied. Moreover, when reporting the results of t-tests the authors should report the t and p values. t values are missing so far, and the authors should report the exact p-values rather than p < 0.05 or p < 0.01.

The discussion should also provide an outlook on the potential use of MIRS and put it into the context of other methods of neuromodulation such as optogenetics and electrical stimulation. So far, the advantages, disadvantages and possible applications are not well explained. What is possible now that the mechanisms of MIRS is better understood?

*Reviewer #3 (Recommendations for the authors):*

This is a study that uses a mid-infrared wavelength light (8.6 um) to induce changes in neuronal response in the pretectal nucleus and in associated optokinetic pursuit eye movements in pigeons. Understanding this manuscript was quite challenging. The study as written lacks a clear hypothesis and experiments lack rationale. No prediction is presented regarding what effect MIRS should have on pretectal neurons or on behavior. Methods are presented without rationale, resulting in what appears to be poor experimental design. In general, the optical stimulation paradigm has a very poor spatiotemporal resolution, making it unlikely to lead to interesting, interpretable results regarding neural circuitry. The optical stimulation produces other concerning effects (e.g. massive sensorimotor responses) for which there are no controls. The rationale for comparing MIRS with electrical stimulation is also obscure. The abstract mentions effect of MIRS can depend on 'ongoing sensory responsiveness in awake animals' but it is not clear what measure is used for sensory responsiveness. The figures are poorly explained. It is unclear why modeling of K^+^ channels is included in this study; it seems like a different topic, making the presentation even more incoherent. Overall, this manuscript lacks rationale, coherence, and direction. A strong suggestion is to (1) describe and understand, at minimum, the spatiotemporal nature of the optical stimulation, (2) make some circuit diagrams regarding what the expected outcomes are of MIRS stimulation on the prectectum, and (3) focus on the neuronal and behavioral portion of this study. Given the primary author's previous strong publications, this should be achievable.

Hypothesis: It would be helpful to state clearly what the hypothesis is and what are the predicted outcomes of stimulating with MIRS in the pretectum. For example, the authors describe that application of MIRS in the pretectum induces an increase in pursuit eye movement velocity in the T-N direction but not in the N-T direction. Does 'facilitating pursuit velocity' mean better behavioral performance? But it is not clear what the significance of this result is. What is the neural circuit underlying OKN and what is the proposed effect of MIRS on this circuit?

Methods: There are many questions about the experimental paradigm that are not addressed. These include:

– why 8.6um wavelength? what is a 'mark of 2 us and a space of 3us'? why use this paradigm? why use a 600um fiber, this is quite large size fiber for a small pigeon pretectum? why 80mW power, this also seems high.

– why 120 seconds (is this a mistake?), this seems quite a long stimulation period, likely to produce non-specific and difficult to interpret effects. For example, typical electrical stimulation might be in the range of 10s or 100s of milliseconds, so 120 seconds is tremendously long. Even optical stimulation, whether it is optogenetic, near infrared (800-900nm, ~1800nm, or mid infrared 5.6um) are usually delivered in briefer sub-second to few seconds periods of time. Why is a 10 min recovery period needed, this is also a very long time.

– The distance between electrode and fiber is 850um. Will such a distance induce a Becqueral effect for 5.8um wavelength light?

– What are the 'massive sensorimotor responses'. This is quite concerning as it suggests there are large multi-circuit effects that are not controlled for.

Results: MIRS

– If the grating motion is 8deg/sec but eye movement velocity is 4.88deg/sec, this is good tracking?

– When the MIRS was turned on, it evoked 'fasted their pursuit performances'. I presume it means the eye movements were faster? Is facilitating pursuit velocity considered an enhanced performance? It is better? Shouldn't an improved response be one that follows the grating better?

– If nasal to temporal pursuit tracking was less effective, what does this indicate about effect on the circuit?

Results: Electrical stimulation

– 120 sec is an extremely long stimulation period. Current would reach every part of the brain.

– ES effectively deflected eye movements to pursue toward the T-N direction: does this mean it interrupted the eye movements? Changed them to T-N direction?

Figures are low resolution

Figure 1: quite hard to understand

– D,F: what are the top and bottom traces, if this a behavioral trace, what does the Before trace look like? Why does the top one look so erratic? Electrophysiology recording trace?

– E,G: what are the 6 graphs? is each dot one animal or one session or one trial?

Figure 3: what is the significance of the positive and negative modulation by MIRS?

Authors should check and review extensively for improvements to the use of English.

---

## [Author Response]

[Editors’ note: The authors appealed the original decision. What follows is the authors’ response to the first round of review.]

Reviewer #1 (Recommendations for the authors):Recent studies suggest that midinfrared stimulation (MIRS) could be a potential neuromodulation strategy, which exerts nonthermal and reversible effect on neuronal signaling and animal behavior. However, there is no direct evidence showing the effect of MIRS on neuronal firing in awake behaving animals. In this study, the authors examined the alternations of neuronal spiking activities in response to MIRS and the association between these alternations with behavioral performance (eye movement in pigeon) in vivo, and compared the modulatory effects of MIRS with electrical stimulation (similar to that with high-frequency deep brain stimulation). With simultaneous extracellular single-unit recording and electrooculogram recording, the authors demonstrated that MIRS with a wavelength of 8.6 μm produced reversible gain modulation on neuronal responsiveness and behavioral performance. They further dissected the underlying potential mechanism using molecular dynamics (MD) simulations and showed that MIRS with specific wavelength could cause carbonyl groups (-C=O) at the ion selectivity filter of K^+^ channel to resonate with the mid-infrared light, leading to the gain modulation of neuronal responsiveness and changes in eye movement. Overall, the electrophysiological results are of high quality and provide strong evidence showing the effect of MIRS on neuronal signaling and sensorimotor behavior. The MD results also provide insight into molecular mechanisms (K^+^ channels) underlying these modulatory effects. However, a weakness of the manuscript is that the methods and results are not well described.1. Please provide detailed information about MIRS laser source and its application. There is no clear description about MIRS parameters. The type of infrared fiber is IRF-S-9 (the diameter should be 9 μm), but the diameter is described as 600 μm.

Thank the reviewer for the comment. We apologize for the error in the prior submission. We have corrected and added detailed information about MIRS laser source and its application in our revised Methods.

2. Please provide the pulse width and repetition frequency of MIRS. The specific parameters (including continue wave or pulse train) should be provided clearly, these parameters will all impact the MIRS effect.

We agreed with the reviewer that the stimulation parameters could impact the MIRS effect. We have reworded the methods with detailed information about MIRS parameters: “The pulse train of MIRS was applied for 30 or 120 seconds with the pulse width of 2 μs, the repetition frequency of 200 kHz, and the duty cycle was 40%. For MIRS experiments tested in the study, the average output powers at the tip of fiber were 2.5, 5, 10 or 20 mW. The stability of output powers within 120 seconds was better than ±0.5%, and measured by a MIR detector (NOVA II-3A, Ophir, Israel).” These detailed parameters were also be provided clearly in Figure 1C, comparing with the protocol of electrical stimulation. We also included new experiments to test how the parameters impact MIRS effects in Figure 4 in the revised manuscript.

3. It would be nice to have a schematic drawing of the experimental configurations, particularly the relative position of the MIRS fiber and the recorded brain region. The authors stated that the distance between the MIRS fiber tip and the electrode recording site was kept ~850 µm. The fiber was inserted vertically (what's the depth?) into the brain tissue. Was the nLM region in the light path?

We thank the reviewer for this instructive comment. To follow the spirit of the reviewer’s suggestion, we have restructured a new Figure 1 to better advantage in introducing our study. The figure includes a schematic drawing of the experimental configurations, and raw EOG and action potential traces to show details during each OKN eye movement. We listed the parameters of MIRS and ES, the retina-nLM-cerebellum circuit involving the OKN eye movement generation, and the information of marked recording sites and MIRS fiber tip sites in the pretectal nLM cross brain sections in our study. We think that most readers would benefit from the orientation in the new Figure 1.

4. What does the "80 mW" represent? the average power? The peak power? Had the authors measured the real output power at the tip of MIRS fiber? What is the estimated power density at the recorded brain region? At least, the power density in air should be given. What was the temperature rise at the fiber tip? Was there any tissue damage? It would help if the authors provided histochemical examination of the brain tissues before and after MIRS, including the nLM region and the fiber tip region. What are the changes in neuronal spiking activity near the fiber tip < 100 μm?

We thank the reviewer for pointing this out. We have corrected and added the detailed information about MIRS output powers. We have also measured profiles of the mid-infrared light spot at a working distance of 850 μm to the fiber tip. The information was included in the revised methods.

A recent research measured the temperature change in the extracellular fluid at different distances from the same laser fiber tip as we used in this study (Tan, et al., 2022). Their results showed that the temperature rise was less than 2℃ within ~1000 μm from the fiber tip when a 10 mW or 20 mW MIRS used. In our experiments, to avoid a collision between the vertical MIRS fiber and horizontal recording electrodes, the recorded nLM neurons located within 848±351 μm below the fiber tip (mean ± SD, n=15, Figure 1 E). Unfortunately, there was no recorded neuron confirmed near the fiber tip <100 μm in our study. Meanwhile, the MIRS effects on behavioral and neuronal responses were revisable after 120 s MIRS irradiation in our observations. Thus, even MIRS could cause limited local thermal heat in the study, the thermal change might have little chance to cause thermal damages in brain tissues. We have rewritten the Discussion section to include these interpretations.

5. Why was the pursuit velocity (but not the duration and distance) altered by MIRS? What's the corresponding relationship between the firing frequency of nLM neurons and the eye movement parameters? How was the pursuit eye velocity analyzed? Please describe these parameters (velocity, duration and distance) and provide details about their measurements. Moreover, the authors performed data analysis of nLM neuron spiking activities within a time window from 0-600 ms after the onset of pursuit. It should be stated clearly in Figure 1 where the onset of pursuit was. The baseline firing activity without visual stimulation should be measured. Were there any effects on baseline activity upon MIRS application? From Figure 3C, it is clear that the firing frequency during the rapid eye resetting phase (i.e. 300 ms before the onset of pursuit) was altered with MIRS. In Figure 3E, the MIRS effect was not reversible? Spiking activity during the rapid phase (before the pursuit onset) was still high when the MIRS was turned off.

Thank the reviewer for the comment. As s/he suggested, we have labeled the related information clearly in the new Figure 1B, including the onset and offset of pursuit. We also have added details about how to measure the amplitude and duration of pursuit, and define pursuit eye velocity as amplitude divided by duration of pursuit in each OKN in. We believe the description of parameters of OKN would better service to orientate readers.

The pretectal nLM receives a direct projection from retina and works as a crucial motion detector to process horizontal visual information, especially in a large visual filed. nLM neurons are sensitive to direction and speed of visual motion. Their firing rates coded the motion information with fine direction tuning curves and speed tuning curves. Then the visual motion information was transferred to the vestibular cerebellum and the ocular cerebellum. They could guide which direction and how fast eyes need to track. So, during the application of MIRS or ES in the nLM, any effective changes in neuronal firing rates could change neuronal coding information, which encoded a “fiction” of visual motion to alter pursuit velocity. We verified MIRS effects on neuronal activity are in a manner of sensory inputs dependent, instead of the ongoing neuronal firing frequency. We measured the effects under two well-controlled conditions in the revised results. First, we tested the “baseline” firing activity while pigeons viewing still grating. It was lacking of visual responses to grating motion in neuronal firing rates. During MIRS, neuronal activities to still grating failed to show any significant changes (Figure 4B, mean ± SEM, -4.15% ± 3.87%, n=6 neurons, p=0.37, two-sided Wilcoxon rank sum test). Second, We confirmed this was true for the subset of cells with similar evoked firing rates preceding MIRS but to different directional motions (Figure 4C). Group 1 preferred nLM cells reached a mean firing rate of 28.17±1.47 spike/s (mean ± SEM, n=7 neurons) during grating motion in their preferred direction. Group 2 null nLM cells reached a mean firing rate of 27.67±1.47 spike/s (mean ± SEM, n=7 neurons) for motion in their null direction. The firing rate range in each group was not statistically different from one another (p=0.80, two-sided Wilcoxon rank sum test), varying from ~22 to ~36 spikes/s. When we compared MIRS effects on both groups, preferred cells showed a larger evoked response (p=0.02, two-sided Wilcoxon signed-rank test), while null cells showed more significant inhibition (p=0.02, two-sided Wilcoxon signed-rank test). These results have consistently supported that MIRS exerts gain modulation of neuronal signals, in a manner that is itself sensory responses dependent.

6. What do the gray bars in Figure 1. D-F stand for?

The gray bars in Figure 1. D-F of prior submission (now the data presented in Figure 1—figure supplement1) used to stand for time windows of 7.5 seconds to show MIRS and ES effects on eye movements. We have restructured the related information in new Figure 2 and Figure 1—figure supplement1.

7. There is inconsistency between results description and Figure 2A. No significant difference between MIRS and Ctrl group?

We have rewritten this part to describe the statistic results in detail, and labeled significant differences in figure 2.

8. The statistic results are inaccurate. Are they significantly different in each condition?

We thank the reviewer for the comment, again. We have run statistical analysis on ES effects in each grating motion condition. The details about statistics were added in results.

9. Should the baselines be aligned in Figure 4A-B? Please describe the statistical results in detail (Paired or independent Student's t-test, repeated-measures ANOVA).

To analyze the MIRS effect on neuronal responses, we collected neuronal activities before, during and after MIRS for about 120 seconds, aligned action potentials to the onset of pursuit, and smoothed by a Gaussian Kernel filter with an *h* value of 25 ms. For each neuron, we computed average firing rate as a function of the 600 ms pursuit time (an example neuron in new Figure 3A and B), and compared its visual responses before and during MIRS. We have reworded the description and included statistical results in detail in the revised manuscript, including the example neuron’s data, and the population neurons’ data.

10. The measurement of the extracellular spike waveforms in Figure S3 is not appropriate. The duration between the trough and the peak (i.e. peak to peak duration) should be used.

We thank the reviewer for the suggestion. Spike width is most commonly measured as the width at half-maximal spike amplitude (Bean, 2007). It is clearer to define the half-height width in intracellular spike waveforms because of a larger depolarization and repolarization. It might be a bit challenging to use the same definition in action potential of extracellular recording, because spikes could be positive-negative or negative-positive waveforms for distinct units. To the concern, we have carefully combined two methods to measure the extracellular spike waveforms in the revised manuscript (Figure 6—figure supplement 1), the full width at half maximum (FWHM) and the duration of the negative phase of the individual spike waveform (spike duration). Both methods were previously used to measure the spike duration of extracellular action potentials (Matsumura, et al., 1997; Takakusaki, et al., 1997; Tattersall, et al., 2014).

11. "Finger" in this manuscript should be changed to "Fingerprint peak".

We have changed the “Finger” to “Fingerprint peak”.

12. From a chemical point of view, the -C=O group only has an intrinsic stretching vibration mode between carbon and oxygen atoms at a frequency of ~53 THz (not 34.88 THz). Although there is an absorption peak of the group at 34.88 THz calculated from the MD simulation, the source of this peak should be explained.

Thank the reviewer for his/her suggestions. Indeed, the stretching vibration frequency of a single carbonyl (-C=O) group should be at ~53 THz, and similarly the corresponding peak ~53.89 THz is also presented in our diagram. However, for the protein secondary structure, the in-plane bending vibration of the -N-C=O groups of the amide VI band corresponds to a low frequency vibration ~19.0 THz, and the -C-N stretching and -N-H in-plane bending correspond to 39.0 THz. (Kauppinen, et al., 1981; Byler and Susi, 1986). It is therefore important to note that the low-frequency oscillation mode corresponding to the frequency 34.88 THz involves almost all of the -C=O groups (similar to -C=O-N-H) in the filter structure vibrating in an in-plane bending manner. Thus, in the MD simulation, data showed two remarkable absorption spectra of K^+^ channels located between 33 to 37 THz, and 50 to 55 THz (Figure 5A). The fingerprint peaks are around ~53 THz and ~35 Thz, just out of the strong absorption spectral ranges of both water molecules (Heyden, et al., 2010) and Na^+^ channels.

13. In this study, the total time of MD simulations is ~10 ns, which is too short for a biological system. It is important to confirm that the MD results are independent of the simulation time.

We thank the reviewer for his/her comment. Action potentials are generated and released in milliseconds, with approximately ~10^7^ ions permeating across the membrane. These ions are passed sequentially through the filter region, so the ion permeation can be studied on the level of single ion. A prior study has set a ~10 ns permeation time for a single ion across the filter when there was no additional traction force (Kopec, et al., 2018). In our work, we also simulated the permeability of a single ion, with an additional traction force constant of 2000 kJ/mol/nm^2^ applied. The permeation time required was set to ~1.5 ns for K^+^ ions and 0.75 ns for Na^+^ ions, at a pulling rate to 0.001 nm/ps. The SMD produced a continuous conformation along the z-direction of the center line of ion channel. The PMF was sampled along the corresponding path. We have reworded and made it clearer that we have studied a single ion permeation in our simulation.

14. The field strength applied is 2.5 V/nm, too strong? Is this strength critical for the gain modulation?

Thank the reviewer for the comment. The reviewer raises an excellent point and we have further conducted computational simulations to test energy potentials as a function of the field strength. We varied the field strength from 0~2.5 V/nm to simulate dynamics of biological systems. The field strength applied was referred to prior studies (Zhu, et al., 2019; Liu, et al., 2021). The new simulation found that MIRS increased the PMF ratio of potassium ions by 1.15 to 1.4 folds at the exit site of K^+^ channels when the strength field increased from 1.0 to 2.5 V/nm (Figure 5D). At the same time, the ratio of Na^+^ channels stayed near 1 at the exit site under different strength fields (Figure 5E). The simulation showed that the potential energy change of potassium ions through the filter became enlarged dramatically by the strength.

15. The filter region of ion channel should be "selectivity filter", not "selective filter".

Thank for pointing the typo. we have corrected “selective filter” to “selectivity filter” in our revised manuscript.

16. PMF is the key data to indirectly support the conclusion that MIRS could preferentially enhance potassium permeability through K^+^ channels to alter action potential generation, but it is not a direct measurement of the change in ion flux. It is important to confirm whether changes in PMF have an effect on K^+^ and Na^+^ currents. What is the direct effect of MIRS on ion currents?

We thank the reviewer for his/her comment and respectfully disagree with the opinion. The PMF map reflects the permeation potential of individual ions along the filter region of ion channels (Bernèche and Roux, 2001; Zachariae, et al., 2014). It can be calculated from molecular dynamics simulations using umbrella sampling. Action potentials are generated and released in milliseconds, with approximately ~10^7^ ions permeating across the membrane. So we think the PMF analysis also indeed directly supports how MIRS could enhance potassium permeability through K^+^ channels to alter action potential generation. It is similar to the ion flux, but in the level of individual ions. Meanwhile, MD simulations also showed that MIRS-induced resonance vibration of the carbonyl groups at these selectivity filter increases the efficiency of the filter and thus the permeability of K^+^ ions (Liu et.al 2021). Another research found that MIRS (with the same frequency of 34.88 THz as our study) can accelerate the rate of ion permeation (i.e. super-permeation) in K^+^ channels for the murine-derived auditory neuron, but had no significant effect on Na^+^ channels (Tan, et al., 2022). We set the cell membrane by a phospholipid bilayer, and the ion channels by intact proteins in our simulations. Supporting by computational resources, we studied the permeation of a single ion through the filter region. We think that the MIRS effects on permeations of multiple ions in sequence should be the same pattern.

17. Can the authors explain the physical mechanisms for the rightward shift of PMF for Na^+^ channels.

We thank the reviewer for his/her careful reading and deep thinking on our data. The PMF calculated by molecular dynamics represents the average free energy (P) change of the system. The mean free energy is introduced in chemical thermodynamics research to determine the direction in which the reflective chemical process proceeds. Generally, *ΔP = ΔH-TΔS*, *T* is the absolute temperature, *ΔS, ΔH* is the change of entropy and enthalpy of the system. If *ΔP* is smaller than 0, it reflects the reactions can be spontaneous. As shown in Figure 5, the free energy of Na^+^ channels is enhanced at the beginning by MIRS. This is attributed to the action of the -OH group, where 34.88 THz corresponds to the mode of action of the -COOH and -OH groups jointly involved in the filter vibration. Thus the chemical potential of Na^+^ channels is first enhanced by the modulation of the -OH groups and then decreased by the modulation of the -COOH groups. As a result, the free energy of the first half is elevated in the presence of the -COOH groups, while the permeation of Na^+^ in the second half is regulated by the -OH vibrations. Thus, it is difficult for Na^+^ to permeate effectively through its filter. All these physical mechanisms could cause the rightward shift of PMF for Na^+^ channels.

18. What is the dependence of PMF and ion currents on MIRS intensity?

Thanks for the comment. A recent work found that MIRS with 34.88 THz can accelerate the ion currents (Tan, et al., 2022).Their simulation demonstrated the ion current is correlated with MIRS intensity when the intensity increased from 0.5 to 2.5 V/nm. In our simulations, we found that the PMF of potassium ions changed in a similar pattern. Our results showed the PMF ratio of K^+^ channels increased by 1.15 to 1.4 folds at the exit site around ξ = 1.1 nm when the MIRS strength field increased from 1.0 to 2.5 V/nm (Figure 5) We have added these results in our revised manuscript.

19. What is the direction of the mid-infrared light? From Figure 5B and C, the direction is parallel to channel pores?

We thank the reviewer for pointing out the importance of the light orientation. In our simulated system, the infrared light is applied in a direction perpendicular to the membrane plane. Due to the symmetry of the channel, the tangential forces of the non-perpendicular light are also symmetrical forces that can be cancelled in the horizontal plane. Thus the effect of light works actually in the direction of the z-axis of the Cartesian coordinate system, along the ion channel center line. We have added the information in the revised manuscript.

Reviewer #2 (Recommendations for the authors):In the manuscript Xiao et al. investigate the impact of mid-infrared stimulation (MIRS) on neuronal activity in awake behaving pigeons. Earlier studies have found that MIRS can excite neuronal responses, but other studies also found inhibition of neuronal firing. In their paper, Xiao et al. investigate the mechanism of MIRS that might explain these diverse findings as reported in the literature so far. They investigated the impact of MIRS and electrical stimulation on neuronal signals as well as on the behavioral performance of pigeons during visually guided eye movements. They found that, unlike electrical stimulation, MIRS rather resulted in gain modulation of firing activity. Therefore, MIRS increased firing rates of highly active neurons, while it reduced the firing of less active cells. Likewise, pursuit eye movements were facilitated by MIRS. Moreover, Xiao et al. explained their experimental findings using computational methods to simulate molecular effects which suggested enhanced permeability of potassium channels.The manuscript from Xiao et al. combines different methods including neurophysiology, behavior and computational modelling to comprehensively investigate the mechanism behind MIRS. Their electrophysiological findings are congruent to their behavioral results supporting the idea that MIRS results in gain modulation. Although this study was performed in pigeons, the findings could be of interest not just for avian researchers but for the whole field of neuroscience. This is the case as they shed light on the mechanisms of MIRS, so that it can be used more purposefully in future as their findings suggest that it is not inhibitory or excitatory per se but results in gain modulation.However, the introduction and/or discussion is missing a section explaining why the effects of MIRS were investigated in pigeons. While the use of diverse model organisms is desirable, the authors motivate their study primarily with clinical applications and say that human research could benefit from MIRS. It is so far not completely clear why the model organism and the specific brain area were chosen in this study. To determine the generalizability of their findings, control experiments in other brain regions or species would be desirable.Moreover, the authors say that in the clinical setting there is the need for neuromodulation methods that, without genetic manipulation, have the potential for neuronal inhibition and excitation. In parts of the manuscript the authors create the impression that this is possible with MIRS. However, based on their findings, the effect of MIRS is not directly controllable, as it varies with the strength of sensory input/neuronal activity. In their experiment the authors were able to control the sensory input into the cell population by varying the visual gratings. However, this is easy for sensory systems and becomes more complicated in higher associative areas. How well could the effect of MIRS be controlled in these areas?

We thank the reviewer for his/her supportive comments. It is rewarding to see a review starts with this kind of comment. The purpose of our study is to reveal the MIRS effects on neuronal and behavioral responses in pigeons. As the reviewer pointed out, we have noticed it might mislead readers to understand our conclusions in the prior version. Now in the revised manuscript, we have restructured and rewritten our description. We made a straightforward presentation and focused on showing results under the context of the sensorimotor transformation. We added new statements explaining why the effects of MIRS were investigated in pigeons in both introduction and Discussion sections. We also added new control experiments to test MIRS effects on pursuit of OKN when MIRS fiber applied in other brain regions. We believe these improvements (suggested by the reviewer) should better service readers to understand MIRS effects in the behavioral and neuronal levels.

The method section is missing important information. For example, the statistics and surgical procedures are not explained. In some other sections, the authors refer to other publications, which makes it hard for the reader to fully understand or assess the procedures.

We thank the reviewer for the comment. We have realized that we had not fully presented our data and stimulation protocol in detail in the prior submission. We do take the responsibility. In revised manuscript, we rewrote the methods, including details about surgery performances, the stimulation protocols, the extracellular action potential recording protocol and the brain tissue processing protocol. We also added a new paragraph about statistics. Now the revised methods are ~2700 words, adding about 1000 words. We believe the new version with detailed information will make our research easier to fully understand and assess the procedures.

The stimulation protocol is also a cause for concern. Testing different stimulation protocols and their effects on neuronal activity would be necessary to fully characterize the effects of MIRS. So far only one protocol was used and it is not made clear why.

We agree with the reviewer. One MIRS protocol could provide effective evidence to show that MIRS indeed modulates behavioral and neuronal responses, but it did not fully characterize the MIRS effects. As the reviewer suggested, we jointly varied visual and infrared stimulation parameters and examined how behavioral and neuronal responses changed. We examined how the strength of sensory input, the output power of MIRS stimulation, and the duration of MIRS irradiation affected responses and included the new results in the revised manuscript.

In the Results section it is sometimes hard to follow the statistical procedures. It is not always indicated what the number of subjects was, which test was performed and why, and whether a within or between subjects design was applied. Moreover, when reporting the results of t-tests the authors should report the t and p values. t values are missing so far, and the authors should report the exact p-values rather than p < 0.05 or p < 0.01.

Thank the review for the comment. As s/he suggested, we have added a new paragraph about statistical procedures in methods, and all related statistical information in the results.

The discussion should also provide an outlook on the potential use of MIRS and put it into the context of other methods of neuromodulation such as optogenetics and electrical stimulation. So far, the advantages, disadvantages and possible applications are not well explained. What is possible now that the mechanisms of MIRS is better understood?

Thank the reviewer for the suggestion. We realized that we had not focused sharply on the effect of MIRS on modulation of behavioral and neuronal responses in animals as we should have. We have reworded the last section in the discussion and try to put MIRS in the context of other neuromodulation methods of optogenetic and electrical stimulation. We think the new manuscript could work better to discuss the mechanisms of MIRS, including a possible topic of future research.

Reviewer #3 (Recommendations for the authors):This is a study that uses a mid-infrared wavelength light (8.6 um) to induce changes in neuronal response in the pretectal nucleus and in associated optokinetic pursuit eye movements in pigeons. Understanding this manuscript was quite challenging. The study as written lacks a clear hypothesis and experiments lack rationale. No prediction is presented regarding what effect MIRS should have on pretectal neurons or on behavior. Methods are presented without rationale, resulting in what appears to be poor experimental design. In general, the optical stimulation paradigm has a very poor spatiotemporal resolution, making it unlikely to lead to interesting, interpretable results regarding neural circuitry. The optical stimulation produces other concerning effects (e.g. massive sensorimotor responses) for which there are no controls. The rationale for comparing MIRS with electrical stimulation is also obscure. The abstract mentions effect of MIRS can depend on 'ongoing sensory responsiveness in awake animals' but it is not clear what measure is used for sensory responsiveness. The figures are poorly explained. It is unclear why modeling of K^+^ channels is included in this study; it seems like a different topic, making the presentation even more incoherent. Overall, this manuscript lacks rationale, coherence, and direction. A strong suggestion is to (1) describe and understand, at minimum, the spatiotemporal nature of the optical stimulation, (2) make some circuit diagrams regarding what the expected outcomes are of MIRS stimulation on the prectectum, and (3) focus on the neuronal and behavioral portion of this study. Given the primary author's previous strong publications, this should be achievable.

The paragraph listed three major concerns and we would like to respond to them in turn:

As the reviewer suggested, we have restructured and rewritten our description about MIRS stimulation in detail. We added new statements explaining why the effects of MIRS were investigated in the nLM of pigeons. We also introduced profile of the MIRS and added new control experiments to test the spatial specificity of MIRS effects on pursuit of OKN. We believe the new version with detailed information will make our research easier to fully understand and assess the procedures.We thank the reviewer for his/her helpful suggestion. To follow the spirit of the reviewer and Reviewer #1’s suggestion, we have tried to present a new Figure 1 to better advantage in introducing our study. Now the new Figure 1D presents the retina-nLM-cerebellum circuit involving the eye movement generation. In the circuit, we outlined firing responses of nLM neurons as a function of visual inputs. And we also summarized and compared the MIRS and ES effects on behavioral and neuronal responses in Figure 6. We think that most readers will benefit from the orientation that is provided by Figure 1 and the statement related in the revised manuscript.We respectfully disagree with the opinion to focus on the neuronal and behavioral portion. We have tried to correct that impression through a complete restructuring of the paper and a large amount of rewriting. We think that the paper is now much more clearly focused on the novel issues it addresses. We would ask the reviewer to re-read the paper with an open mind. We hope that s/he will find that the potential molecular mechanism studied by ion channels simulation could serve as an important piece of puzzle to fill in the whole figure of the mechanism of MIRS.

Hypothesis: It would be helpful to state clearly what the hypothesis is and what are the predicted outcomes of stimulating with MIRS in the pretectum. For example, the authors describe that application of MIRS in the pretectum induces an increase in pursuit eye movement velocity in the T-N direction but not in the N-T direction. Does 'facilitating pursuit velocity' mean better behavioral performance? But it is not clear what the significance of this result is. What is the neural circuit underlying OKN and what is the proposed effect of MIRS on this circuit?

We thank the reviewer for the suggestion. The pretectal nLM receives a direct projection from retina and works as a key motion detector to process horizontal visual information, especially in a large filed. nLM neurons are sensitive to motion direction and speed. Their firing frequency coded the motion information including direction and speed. Once these visual motion information were transferred to the vestibular cerebellum and the oculomotor cerebellum, they could guide which direction and how fast eyes need to move. So, during the manipulation of MIRS or ES, any effective changes in neuronal activity could code a fiction of grating motion and cause pursuit velocity altered. The related description was clarified in the revised manuscript.

Methods: There are many questions about the experimental paradigm that are not addressed. These include:– why 8.6um wavelength? what is a 'mark of 2 us and a space of 3us'? why use this paradigm? why use a 600um fiber, this is quite large size fiber for a small pigeon pretectum? why 80mW power, this also seems high.

We have corrected and added detailed information about MIRS in our revised version.

We agree with the reviewer that a 600 μm optical fiber is a large size, but the laser could have a good chance to irradiate the pretectal nLM with a width of ~1000 μm. We measured profiles of the light spot when the mid-infrared laser passed through the coupler and optical fiber, and travelled 850 μm distance to the test surface. The working distance of 850 μm is the average distance between the fiber tip and recorded neurons in our experiments (Figure 1E). We used the MIR detector covered by a tinfoil with a 100 μm diameter hole to scan the intensity of the light, with a scanning resolution of 100 μm (Figure 4—figure supplement 1A and B). The spot diameter was ~1000 μm when the laser light intensity was reduced to 1/e^2^ of the peak in our setting. Thus, the infrared fiber with a diameter of 600 μm could have a good probability that the nLM neurons located in the light path of MIRS. Meanwhile, the MIRS effects on behavioral and neuronal responses were revisable after MIRS irradiation in our observations.

– why 120 seconds (is this a mistake?), this seems quite a long stimulation period, likely to produce non-specific and difficult to interpret effects. For example, typical electrical stimulation might be in the range of 10s or 100s of milliseconds, so 120 seconds is tremendously long. Even optical stimulation, whether it is optogenetic, near infrared (800-900nm, ~1800nm, or mid infrared 5.6um) are usually delivered in briefer sub-second to few seconds periods of time.

Thank the reviewer for this comment. When we designed the stimulation protocol, we have carefully checked related researches on MIRS and ES in animals. We have confirmed that MIRS durations applied from tens to hundreds of seconds (references listed below 1-3). And the DBS has been used as a chronical stimulation in animals lasting minutes to hours a day (listed below 4-6). Referred by these prior researches, we have set the MIRS and ES period of 120 s in our experimental protocol. The ES parameters are similar to DBS used in mice, with current of 200 μA, frequency of 133 Hz, and pulse width of 100 μs.

“The duration of MIRS varied from 10 to 200 s covering the whole periods of testing for its effects.” (Liu, et al., 2021)“For animals in MIM group, the MIR irradiation was turned on (by a sound-free electronic shutter) during the engagement time window (which was 20–35 s depending on the random inter-trial interval, similar to the MIM applications in the two-photon imaging experiments).” (Zhang, et al., 2021)“MET currents were recorded in one-minute cycles, with THM applied at 10 s and then switched off at 40 s” (Tan, et al., 2022)“Mice were chronically stimulated for 25 days for 7 hours per day with weekends off, using 50 μA at a frequency of 130 Hz and pulse width of 90 μs“ (Mann, et al., 2018)“Animals in DBS groups received 1 hour DBS daily for 14 consecutive days. The DBS was biphasic rectangular pulses (130 Hz, 60 μs pulse duration, 68.62 ± 13.87 μA) (Hao, et al., 2015)The DBS stimulation was applied with parameters of electrical currents (60 μs, 120 μA, 130 Hz) for 2 minutes. (Valverde, et al., 2020)

A further possible concern related it could be the function of MIRS periods: short term of tens of seconds versus long term of several minutes. To probe the effect of stimulation duration, we run a new experiment and compared the influence of MIRS applied for 30 s vs. 120 s during 8 deg/s grating motion. We found evoked pursuit velocities did not vary substantially as a function of MIRS duration. When we divided the 120 s duration condition into 4 sequential 30 s segments, we also found that the MIRS duration was not associated with a significant difference in pursuit eye velocity or nLM firing rates. These results were presented in the new Figure 4G-J and listed in results.

Why is a 10 min recovery period needed, this is also a very long time.

Thank the reviewer for this comment. As s/he pointed, a 10 min recovery period could be a long time. As the results showed, MIRS effects can be vanished within a couple of minutes after MIRS turned off. We also tried to set a “shorter” recovery period of 2-3 minutes in the new experiments (Figure 4D and H). Data showed that behavioral and neuronal responses can recovery from MIRS modulations within 2-3 minutes.

– The distance between electrode and fiber is 850um. Will such a distance induce a Becqueral effect for 5.8um wavelength light?

According to a prior research of 5.6 μm wavelength light, the effect of MIRS on action potential decays slowly as the distance between MIRS fiber tip and the recorded cells. MIRS exerted a long-distance effect on AP waveforms. The effective distance was about 1200 to 1500 μm (Liu, et al., 2021). To avoid a collision between vertical MIRS fiber and horizontal recording electrode in our experiments, we collected nLM neurons located about 848±351 μm below the fiber tip (Figure 1E). We also measured the mid-infrared light spot diameter at a working distance of 850 μm, which was about 1000 μm (Figure 4—figure supplement 1A and B). Thus the laser could have a good chance to irradiate the pretectal nLM with a width of ~1000 μm. With these observation, we think most of the tested neurons could have a high chance to be modulated effectively by MIRS.

– What are the 'massive sensorimotor responses'. This is quite concerning as it suggests there are large multi-circuit effects that are not controlled for.

In our prior submission, we have used the word “massive” to state massive asymmetric sensorimotor responses. When pigeons viewed a grating motion of 8 deg/s in the temporo-to-nasal (T-N) direction, they can pursue moving gratings at velocity of 4.78±0.22 deg/s along the T-N direction. Once MIRS was turned on, animals significantly fasted their pursuit velocities to 6.44±0.38 deg/s. Conversely, when pigeons were introduced a grating motion of 8 deg/s in the nasal-to-temporal (N-T) direction, animals tracked grating motion with far less effective pursuit eye movements at velocity of -0.99±0.06 deg/s. When MIRS was turned on, animals again significantly fasted their eye movements to pursue in the N-T direction at tracking velocity of -1.48±0.08 deg/s. There was asymmetry between the N–T and T–N OKN, which has been widely observed in lateral-eyed vertebrates (rabbits: Collewijn, 1969; pigeons: Zolotilina, et al., 1995 rats: Harvey, et al., 1997; mice: Kodama and du Lac, 2016). In the revised manuscript, we have removed the implication and reworded the sentence.

Results: MIRS– If the grating motion is 8deg/sec but eye movement velocity is 4.88deg/sec, this is good tracking?

When the grating motion is 8deg/sec, pigeons can track at 4.88 deg/sec (now mean pursuit velocity is 4.78 deg/sec for 20 pigeons) with a pursuit gain about 0.6. To our best knowledge, it is common and good tracking in pigeons’ eye performances (Gioanni, 1988; Yang, et al., 2008b).

– When the MIRS was turned on, it evoked 'fasted their pursuit performances'. I presume it means the eye movements were faster? Is facilitating pursuit velocity considered an enhanced performance? It is better? Shouldn't an improved response be one that follows the grating better?

Yes, as the reviewer pointed, eye movements were faster during MIRS. Pigeons track the grating motion with faster eye movements in both T-N and N-T directions when grating motion is 8 deg/sec. We have revised the description in a clear way.

– If nasal to temporal pursuit tracking was less effective, what does this indicate about effect on the circuit?

Pursuit tracking in the N-T direction was less effective than in the T-N direction. There was asymmetry between the N–T and T–N OKN, which has been widely observed in lateral-eyed vertebrates (rabbits: Collewijn, 1969; pigeons: Zolotilina, et al., 1995 rats: Harvey, et al., 1997; mice: Kodama and du Lac, 2016). It is an interesting question about neuronal mechanisms underlying the asymmetry. In the nLM, majority of neurons become excited by grating motion in the T-N direction, whereas other neurons are predominantly sensitive to the nasal-to-temporal, or vertical motion. Lesions of the nLM could impair horizontal eye movements, especially in the T-N direction (Gioanni, et al., 1983). Thus, we think, the pretectal nLM is a crucial region in the optokinetic circuit involved.

Results: Electrical stimulation– 120 sec is an extremely long stimulation period. Current would reach every part of the brain.

The DBS has been used as a chronical stimulation in animals lasting minutes to hours a day. Referred by prior researches of DBS, we have set the ES period of 120 s in our experimental protocol. The parameters of ES are similar to the DBS in mice, using 200 μA at a frequency of 133 Hz, and pulse width of 100μs. We have reworded the information and added references in introduction and methods.

– ES effectively deflected eye movements to pursue toward the T-N direction: does this mean it interrupted the eye movements? Changed them to T-N direction?

Yes. When animals were introduced grating motion in the N-T direction, they tracked in that direction by their OKN. When ES used, eye movements pursued toward the T-N direction. In the nLM, majority of neurons become excited by grating motion in the T-N direction, whereas other neurons are predominantly sensitive to the nasal-to-temporal, or vertical motion. Once ES was applied, nLM neurons can be activated to increase firing. The population coding in nLM could report a “fiction” of visual motion towards the T-N direction because of major T-N preferred neurons. Then, the eye movements can be effectively interrupted and changed to pursuit in the T-N direction.

Figures are low resolution

We apologize for it. We improved the resolution of all figures.

Figure 1: quite hard to understand– D,F: what are the top and bottom traces, if this a behavioral trace, what does the Before trace look like? Why does the top one look so erratic? Electrophysiology recording trace?– E,G: what are the 6 graphs? is each dot one animal or one session or one trial?

Thank the reviewer for pointing it out. The Figure 1 used to be densely packed with information of the experimental setup, and behavioral data. We understand that it might cause readers hard to follow and we take responsibility. We have reorganized graphs and added detailed information in the revised manuscript. Inspired by Reviewer #3 and #1’s comments, we now have recreated new figures. A new Figure 1 is to focus on introducing experimental configurations. A new Figure 2 and a new Figure 2—figure supplement 1are to show behavioral data during MIRS and ES with a clearer presentation. We think that most readers will benefit from the orientation of experimental protocols provided in Figure 1 and the focused presentation listed in Figure 2.

Figure 3: what is the significance of the positive and negative modulation by MIRS?

We have added information of statistics in Figure 3 and results.

Authors should check and review extensively for improvements to the use of English.

With the help from a native English speaker, we have carefully revised the manuscript and improved it its readability.

References

1. Bean BP. 2007. The action potential in mammalian central neurons. *Nature Reviews Neuroscience* 8: (6) 451-465. doi: https://doi.org/10.1038/nrn2148.

2. Bernèche S; Roux B. 2001. Energetics of ion conduction through the K^+^ channel. *Nature* 414: (6859) 73-7. doi: https://doi.org/10.1038/35102067.

3. Byler DM; Susi H. 1986. Examination of the secondary structure of proteins by deconvolved FTIR spectra. *Biopolymers: Original Research on Biomolecules* 25: (3) 469-487. doi:

4. Collewijn H. 1969. Changes in visual evoked responses during the fast phase of optokinetic nystagmus in the rabbit. *Vision Res* 9: (7) 803-14. doi: https://doi.org/10.1016/0042-6989(69)90016-9.

5. Gioanni H. 1988. Stabilizing gaze reflexes in the pigeon (Columba livia). *Experimental Brain Research* 69: (3) 567-582. doi: https://doi.org/10.1007/BF00247310.

6. Gioanni H; Rey J; Villalobos J; Richard D; Dalbera A. 1983. Optokinetic nystagmus in the pigeon (Columba livia). II. Role of the pretectal nucleus of the accessory optic system (AOS). *Exp Brain Res* 50: (2-3) 237-47. doi: https://doi.org/10.1007/bf00239188.

7. Hao S; Tang B; Wu Z; Ure K; Sun Y; Tao H; Gao Y; Patel AJ; Curry DJ; Samaco RC; Zoghbi HY; Tang J. 2015. Forniceal deep brain stimulation rescues hippocampal memory in Rett syndrome mice. *Nature* 526: (7573) 430-4. doi: https://doi.org/10.1038/nature15694.

8. Harvey RJ; De'sperati C; Strata P. 1997. The early phase of horizontal optokinetic responses in the pigmented rat and the effects of lesions of the visual cortex. *Vision Res* 37: (12) 1615-25. doi: https://doi.org/10.1016/s0042-6989(96)00292-1.

9. Heyden M; Sun J; Funkner S; Mathias G; Forbert H; Havenith M; Marx D. 2010. Dissecting the THz spectrum of liquid water from first principles via correlations in time and space. *Proc Natl Acad Sci U S A* 107: (27) 12068-73. doi: https://doi.org/10.1073/pnas.0914885107.

10. Kauppinen JK; Moffatt DJ; Mantsch HH; Cameron DG. 1981. Fourier transforms in the computation of self-deconvoluted and first-order derivative spectra of overlapped band contours. *Analytical Chemistry* 53: (9) 1454-1457. doi:

11. Kodama T; Du Lac S. 2016. Adaptive Acceleration of Visually Evoked Smooth Eye Movements in Mice. *The Journal of Neuroscience* 36: (25) 6836-6849. doi: https://doi.org/10.1523/jneurosci.0067-16.2016.

12. Kopec W; Köpfer DA; Vickery ON; Bondarenko AS; Jansen TLC; De Groot BL; Zachariae U. 2018. Direct knock-on of desolvated ions governs strict ion selectivity in K(+) channels. *Nat Chem* 10: (8) 813-820. doi: https://doi.org/10.1038/s41557-018-0105-9.

13. Krauss JK; Lipsman N; Aziz T; Boutet A; Brown P; Chang JW; Davidson B; Grill WM; Hariz MI; Horn A; Schulder M; Mammis A; Tass PA; Volkmann J; Lozano AM. 2021. Technology of deep brain stimulation: current status and future directions. *Nat Rev Neurol* 17: (2) 75-87. doi: https://doi.org/10.1038/s41582-020-00426-z.

14. Liu X; Qiao Z; Chai Y; Zhu Z; Wu K; Ji W; Li D; Xiao Y; Mao L; Chang C; Wen Q; Song B; Shu Y. 2021. Nonthermal and reversible control of neuronal signaling and behavior by midinfrared stimulation. *Proc Natl Acad Sci U S A* 118: (10). doi: https://doi.org/10.1073/pnas.2015685118.

15. Mann A; Gondard E; Tampellini D; Milsted JaT; Marillac D; Hamani C; Kalia SK; Lozano AM. 2018. Chronic deep brain stimulation in an Alzheimer's disease mouse model enhances memory and reduces pathological hallmarks. *Brain Stimul* 11: (2) 435-444. doi: https://doi.org/10.1016/j.brs.2017.11.012.

16. Matsumura M; Watanabe K; Ohye C. 1997. Single-unit activity in the primate nucleus tegmenti pedunculopontinus related to voluntary arm movement. *Neurosci Res* 28: (2) 155-65. doi: https://doi.org/10.1016/s0168-0102(97)00039-4.

17. Takakusaki K; Shiroyama T; Kitai ST. 1997. Two types of cholinergic neurons in the rat tegmental pedunculopontine nucleus: electrophysiological and morphological characterization. *Neuroscience* 79: (4) 1089-109. doi: https://doi.org/10.1016/s0306-4522(97)00019-5.

18. Tan XX; Wu KJ; Liu S; Yuan YF; Chang C; Xiong W. 2022. Minimal-invasive enhancement of auditory perception by terahertz wave modulation. *Nano Research* 15: (6) 5235-5244. doi: https://doi.org/10.1007/s12274-022-4127-7.

19. Tattersall TL; Stratton PG; Coyne TJ; Cook R; Silberstein P; Silburn PA; Windels F; Sah P. 2014. Imagined gait modulates neuronal network dynamics in the human pedunculopontine nucleus. *Nat Neurosci* 17: (3) 449-54. doi: https://doi.org/10.1038/nn.3642.

20. Valverde S; Vandecasteele M; Piette C; Derousseaux W; Gangarossa G; Aristieta Arbelaiz A; Touboul J; Degos B; Venance L. 2020. Deep brain stimulation-guided optogenetic rescue of parkinsonian symptoms. *Nat Commun* 11: (1) 2388. doi: https://doi.org/10.1038/s41467-020-16046-6.

21. Yang Y; Yang Y; Wang SR. 2008b. Neuronal circuitry and discharge patterns controlling eye movements in the pigeon. *J Neurosci* 28: (42) 10772-80. doi: https://doi.org/10.1523/jneurosci.2468-08.2008.

22. Zachariae; Ulrich; Sheldrick; George; M.; Koepfer; David; A.; Gruene; Tim. 2014. Ion permeation in K^+^ channels occurs by direct Coulomb knock-on. *Science*. doi:

23. Zhang J; He Y; Liang S; Liao X; Li T; Qiao Z; Chang C; Jia H; Chen X. 2021. Non-invasive, opsin-free mid-infrared modulation activates cortical neurons and accelerates associative learning. *Nat Commun* 12: (1) 2730. doi: https://doi.org/10.1038/s41467-021-23025-y.

24. Zolotilina EG; Eremina SV; Orlov IV. 1995. Horizontal optokinetic nystagmus in the pigeon during static tilts in the frontal plane. *Neurosci Behav Physiol* 25: (4) 300-6. doi: https://doi.org/10.1007/bf02360041.